# Homozygous *MTAP* deletion in primary human glioblastoma is not associated with elevation of methylthioadenosine

Yasaman Barekatain [1,2,3 ✉], Jeffrey J. Ackroyd[1,3], Victoria C. Yan [1,3], Sunada Khadka[1,2,3], Lin Wang[4], Ko-Chien Chen[2,3], Anton H. Poral[1], Theresa Tran[1], Dimitra K. Georgiou [1], Kenisha Arthur[1], Yu-Hsi Lin[1], Nikunj Satani [1], Elliot S. Ballato [1], Eliot I. Behr [1,2], Ana C. deCarvalho [5], Roel G. W. Verhaak[6], John de Groot[7], Jason T. Huse[8], John M. Asara[9], Raghu Kalluri [2] & Florian L. Muller [1,10 ✉]

Homozygous deletion of methylthioadenosine phosphorylase (*MTAP*) in cancers such as glioblastoma represents a potentially targetable vulnerability. Homozygous *MTAP*-deleted cell lines in culture show elevation of MTAP's substrate metabolite, methylthioadenosine (MTA). High levels of MTA inhibit protein arginine methyltransferase 5 (PRMT5), which sensitizes *MTAP*-deleted cells to PRMT5 and methionine adenosyltransferase 2A (MAT2A) inhibition. While this concept has been extensively corroborated in vitro, the clinical relevance relies on exhibiting significant MTA accumulation in human glioblastoma. In this work, using comprehensive metabolomic profiling, we show that MTA secreted by *MTAP*-deleted cells in vitro results in high levels of extracellular MTA. We further demonstrate that homozygous *MTAP*-deleted primary glioblastoma tumors do not significantly accumulate MTA in vivo due to metabolism of MTA by MTAP-expressing stroma. These findings highlight metabolic discrepancies between in vitro models and primary human tumors that must be considered when developing strategies for precision therapies targeting glioblastoma with homozygous *MTAP* deletion.

[1] Department of Cancer Systems Imaging, The University of Texas MD Anderson Cancer Center, Houston, TX, USA. [2] Department of Cancer Biology, The University of Texas MD Anderson Cancer Center, Houston, TX, USA. [3] MD Anderson UT Health Graduate School of Biomedical Sciences, Houston, TX, USA. [4] Department of Chemistry and Lewis-Sigler Institute for Integrative Genomics, Princeton University, Princeton, NJ, USA. [5] Department of Neurosurgery, Henry Ford Hospital, Detroit, MI, USA. [6] The Jackson Laboratory for Genomic Medicine, Farmington, CT, USA. [7] Department of Neuro-Oncology, The University of Texas MD Anderson Cancer Center, Houston, TX, USA. [8] Department of Pathology, The University of Texas MD Anderson Cancer Center, Houston, TX, USA. [9] Department of Medicine, Harvard Medical School, and Division of Signal Transduction, Beth Israel Deaconess Medical Center, Boston, MA, USA. [10] SPOROS Bioventures, Houston, TX, USA. ✉email: yasbarekat@gmail.com; aettius@aol.com

Genomic deletions of tumor-suppressor genes are prevalent drivers of tumor progression in various cancers, but their therapeutic inaccessibility subverts their potential use via precision oncology. Such deletion events often confer less-obvious genetic vulnerabilities that may be therapeutically exploited by identifying their metabolic consequences[1,2]. Homozygous deletion of CDKN2A/B at the 9p21 chromosome is an early clonal event in tumorigenesis[3] that is homogeneously distributed in glioblastoma multiforme (GBM), pancreatic adenocarcinoma, and lung cancer[4]. Within the context of GBM, homozygous 9p21 deletions are found in around 30–50% of all cases[5]. Of particular interest in developing therapies that capitalize on aberrant tumor metabolism is collateral deletion of the evolutionarily conserved metabolic enzyme methylthioadenosine phosphorylase (encoded by MTAP). Owing to the proximity of MTAP to the CDKN2A tumor-suppressor locus[6,7], co-deletion of MTAP may be observed in 80–90% of all tumors harboring homozygous deletion of CDKN2A[8] (Supplementary Fig. 1). The ubiquity of this event alongside the poor prognosis of cancers such as GBM has urged the development of novel therapies that capitalize on downstream vulnerabilities conferred by MTAP deletion.

In addition to its broader role in polyamine biosynthesis, MTAP is critically involved in the salvage pathways of both methionine and adenine, catalyzing the conversion of methyl-thioadenosine (MTA) to S-methyl-5-thio-D-ribose-1-phosphate (Fig. 1a). Cancers sustaining homozygous MTAP deletions are thus expected to accumulate MTA; extensive in vitro evidence in diverse cancer cell lines supports this logic[9,10] (though methionine and cysteine availability in media also influences MTA levels[11]). Efforts to act on this intriguing metabolic phenotype have identified the inhibitory effect of excess MTA on protein arginine methyltransferase 5 (PRMT5), a key regulator of transcription. PRMT5 exerts its regulatory effects when conjugated with WD repeat-containing protein (WDR77) to generate what is known as the methylosome[12–14]. High levels of MTA act as an endogenous inhibitor of PRMT5 activity, thereby hindering methylosome formation and sensitizing cells to reduced levels of PRMT5 and WDR77. Therapeutic targeting of PRMT5 in homozygous MTAP-deleted cancer cells has thus been considered a promising strategy to prevent methylosome organization, selectively killing cancer cells[9,10,15] (Supplementary Fig. 1). Another approach leverages the relationship between PRMT5 and S-adenosylmethionine (SAM)[9,16,17]. SAM is the natural substrate of PRMT5, and MTA is a substrate-competitive inhibitor of PRMT5. Lowering levels of SAM would thus exacerbate the inhibitory effect of elevated MTA. Accordingly, inhibition of methionine adenosyltransferase 2A (MAT2A), which generates SAM, has also been considered as a potential therapeutic target in homozygous MTAP-deleted cancers[9,16,17] (Supplementary Fig. 1), and this strategy has progressed to an ongoing phase I trial (NCT03435250).

Both of the aforementioned therapeutic approaches are predicated on the presence of exceedingly high MTA levels in MTAP-deleted cancer cells compared to MTAP-intact tissues. While it may seem natural that homozygous MTAP-deleted primary human tumors should also display this phenotype, recent reports identifying this intriguing vulnerability have not measured MTA levels in primary human tumors[9,10,18]. We could not find any previous studies that reported MTA measurements in actual primary human tumors as a function of the MTAP genotype.

In this study, we found that highly elevated MTA levels found in MTAP-deleted glioma cell lines in culture cannot be extrapolated to primary GBMs. Our series of metabolomic studies in cell culture and primary tumors demonstrate that MTA accumulation in culture is most evident extracellularly, reflecting

secretion of MTA by MTAP-deleted cells. Building on this finding in primary tumors, our data strongly suggest that the abundance of non-malignant MTAP wild-type (WT) stromal cells metabolize the secreted MTA from the homozygous MTAP-deleted GBM cells. As the promise of synthetic therapies that are lethal against homozygous MTAP-deleted cancers is dependent on intracellular accumulation of MTA, our data caution against the expedient translation of the MTAP-deletion-targeted precision therapies to the clinic and strongly contend that more research is needed into the fundamental metabolic differences between model systems and primary tumors.

## Results

**Secretion of MTA by MTAP-deleted cells in culture.** By analyzing multiple metabolic profiling studies using independent metabolomic platforms, we observed a discrepancy between reported intracellular MTA levels in homozygous MTAP-deleted cells[9,19–25], even within the same cell lines (Supplementary Fig. 2a, b). This discrepancy can be attributed to the extent to which these studies distinguish between intracellular and extracellular distribution—which depends on how extensively the cells were washed before metabolite extraction. For instance, in the NCI-60 profiling study by Ortmayr et al.[22], which focused exclusively on intracellular metabolites (extensive washing of cell pellets with minimal residual media), there was no statistically significant increase in MTA levels in MTAP-deleted compared to WT cancer cell lines (Supplementary Fig. 2a). However, a collaboration between Metabolon, Inc. and the NCI using the NCI-60 panel[26] did not specify whether the cell pellet was washed and reported a threefold increase in MTA in MTAP-deleted compared to MTAP-WT cell lines (Supplementary Fig. 2b). In contrast to intracellular levels of MTA, re-analysis of one notable study that conducted mass spectrometry (MS) profiling of conditioned media (secreted metabolites) showed >100-fold increases in MTA in MTAP-deleted cell lines compared to intact cell lines (Supplementary Fig. 2c; for comparison with lactate, see Supplementary Fig. 2d)[27]. The micromolar extracellular levels of MTA reported by these studies sharply contrasts the nanomolar to picomolar levels of MTA typically found in conditioned media of MTAP-WT cells[24,28]. Also, direct measurement of MTA secretion from MTAP-deleted cells into culture medium showed a significantly higher rate of secretion from MTAP-deleted cell lines than from MTAP-WT lines[19,29].

To independently investigate the likelihood of MTA's predominant presence in conditioned media from MTAP-deleted cells, we performed mass spectroscopy on cell pellets and corresponding conditioned media (Fig. 1b) from the panel of cell lines with verified MTAP status (Fig. 1c and Supplementary Fig. 3). First, we compared MTA levels from the intracellular and extracellular environments of three MTAP-deleted glioma cell lines (U87, Gli56, and SW1088) with MTAP-WT cell lines (Fig. 1d, e). We observed dramatic increases in MTA levels in conditioned media of MTAP-deleted versus WT cell lines but a much more modest increase of MTA levels in the cell pellets of these respective cell lines. The significant differences in extracellular MTA between MTAP-deleted and WT compared to the minor differences in intracellular MTA levels of the same cells indicate that MTAP-deleted cells secrete MTA (Fig. 1d versus Fig. 1e). To corroborate our findings, we also performed a time-course experiment for MTAP-deleted and MTAP-WT cell lines. Although a time-dependent increase in MTA was apparent in both the cell pellets and conditioned media, the increase was exacerbated in the extracellular profile (Fig. 1f). Comparison of intracellular and extracellular levels of MTA and SAM demonstrated the tight regulation of intracellularly retained versus

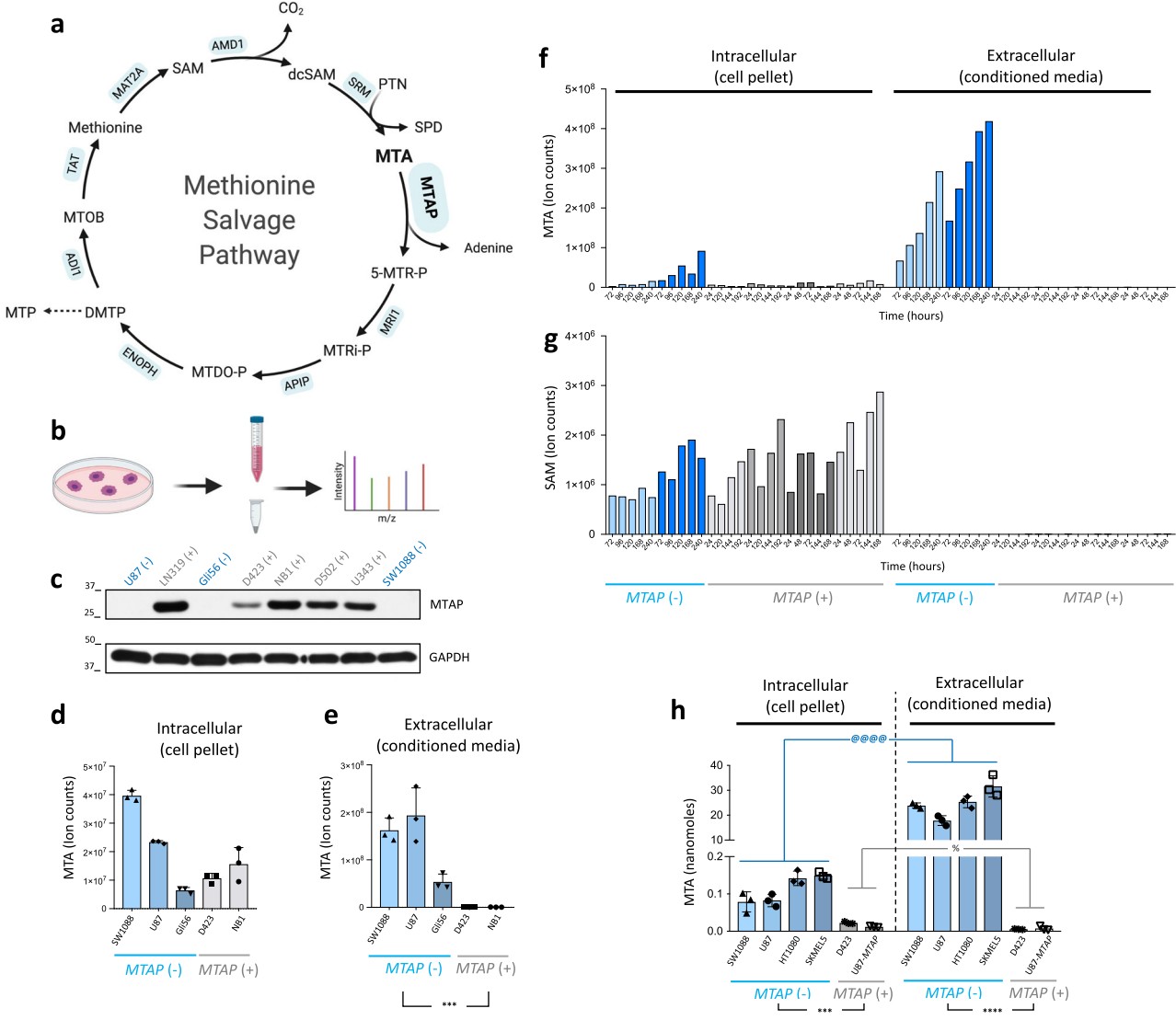

**Fig. 1 Dramatic elevation of MTA in conditioned medium of *MTAP*-deleted cells versus WT coupled with mild intracellular elevation. a** The methionine salvage pathway (based on KEGG[68]) with most genes are housekeeping expressed broadly across cell lines and tissues (DepMap, The Human Protein Atlas). SAM S-adenosylmethionine, dcSAM S-adenosylmethionineamine, MTA methylthioadenosine, SPD spermidine, PTN putrescine, MTOB 4-methylthio-2-oxobutanoate, DMTP 1,2-dihydroxy-5-(methylthio)pent-1-en-3-one, MTP 3-(methylthio)propanoate, MTDO-P 5-(methylthio)-2,3-dioxopentyl-phosphate, MTRi-P S-methyl-5-thio-D-ribulose-1-phosphate, 5-MTR-P methylthioribose-1-phosphate. **b** MTA levels were determined in conditioned media (extracellular) and washed cell pellets (intracellular) of homozygous *MTAP*-deleted (*MTAP*−) and wild-type (*MTAP*+) cells in culture. Color coding reflects the same cell line for pellet/media. **c** *MTAP* status of cell lines confirmed by western blot repeated independently twice. **d, e** Levels of MTA (mean + SD, N = 3 biological replicates) in a panel of glioma cell; washed cell pellet—intracellular (**d**) and conditioned media—extracellular (**e**). Note >200-fold increase in MTA in conditioned media of *MTAP*-deleted versus intact cell lines (***$p = 0.0005$, unpaired two-tailed Student's *t* test with unequal variance), while only marginal elevations are seen for this comparison in the cell pellet. **f, g** Time course of MTA levels in cell pellet and conditioned media (**f**) with SAM shown for comparison (**g**). x-axis: time of harvest after last media change; y-axis: ion counts for each metabolite. Each bar represents one biological replicate. There was a distinct, time-dependent increase in MTA in media of *MTAP*-deleted cells with a modest increase of MTA in the cell pellet. There was an imbalance of MTA in the pellet versus the media, contrasting what was observed with SAM, which is exclusively intracellular. **h** Absolute quantification of MTA in cell pellet and conditioned media (mean +/− SD, N = 3 biological replicates). The amount of MTA recovered from conditioned media of *MTAP*-deleted cells is 200-fold greater than recovered from the cell pellets. In contrast, in *MTAP*-WT cells, the amount of MTA in conditioned media is comparable to that in cell pellet. The modest intracellular accumulation of MTA (sixfold) in the *MTAP*-deleted cells compared to the WT cells is overshadowed by the increase of extracellular MTA. Significant values are indicated as ****$p = 4 \times 10^{-8}$, ***$p = 0.00006$, @@@@$p = 10^{-12}$, and %$p = 0.02$ using multiple *t* test with Bonferroni correction.

secreted metabolites (Fig. 1f versus Fig. 1g). We further measured the absolute amount (nanomoles) of MTA for a paired sample of cell pellets and conditioned media. Figure 1h indicates that the amount of MTA recovered from the conditioned media from *MTAP*-deleted cells cultured for 3 days is approximately 200 times more than MTA recovered from the cell pellets. This is in sharp contrast with the *MTAP*-WT cells, where a comparable amount of MTA was recovered from media (extracellular) and the cell pellet (intracellular). Taken together, our findings agree with previous conclusions [19,29] that homozygous *MTAP*-deleted cells secret MTA resulting in significant extracellular accumulation of MTA.

**No significant elevation of MTA in homozygous *MTAP*-deleted human GBMs**. While elevated MTA levels in vitro clearly define a promising actionable metabolic vulnerability, none of the studies[9,10,15,18] reported on the critical question of whether homozygous *MTAP*-deleted primary human tumors faithfully replicate the elevation of MTA observed in vitro. In other words, does the metabolic vulnerability mediated by homozygous *MTAP* deletion hold in primary human tumors, including GBM? To answer this question directly, using our metabolomic profiling data generated with the BIDMC platform, we compared MTA levels between resected human GBM tumors with verified *MTAP*-deletion status[30–32] (Fig. 2a, b). This is the first study to address *MTAP* deletion's effects on MTA levels in human tumors. Figure 2c shows that, in 17 GBM tumors (HF series), MTA levels do not differ significantly between homozygous *MTAP*-deleted and *MTAP*-intact tumors (~1.4-fold higher median MTA levels in *MTAP*-deleted tumors; $p = 0.20$, unpaired 2-tailed $t$ test with unequal variance). When corrected for loading, the median MTA levels were only 1.2-fold higher in *MTAP*-deleted than in *MTAP*-intact tumors ($p = 0.09$; Fig. 2d). Next, we normalized MTA levels to SAM levels for each tumor to account for the methionine salvage pathway's upregulation (Fig. 2e). Since the levels of SAM were not significantly different between *MTAP*-deleted and *MTAP*-WT cells in culture[15] (data replotted in Supplementary Fig. 4), SAM is a suitable metabolite for normalization purposes. When the MTA levels were corrected to SAM levels in primary human GBM tumors, the difference between *MTAP*-deleted and *MTAP*-intact tumors became decisively nonsignificant (1.1-fold higher median MTA levels in *MTAP*-deleted tumors; $p = 0.24$; Fig. 2e). This result runs contrary to the expected phenotype in homozygous *MTAP*-deleted cells in culture. This nonsignificant MTA elevation pales further compared to outliers driven by specific genetic events that fully recapitulate metabolic data recorded intracellularly in cell culture, such as the >100-fold increase in the levels of 2-hydroxyglutarate (2-HG), driven by the point mutation of *IDH1* (Fig. 2f).

To further confirm the discrepancy in MTA levels in human GBM tumors and cells in culture, we measured MTA levels in a different set of tumors (MDA series) using a different metabolomic platform (Metabolon, Inc.). The *MTAP* status of tumors was confirmed by immunohistochemistry using a validated MTAP antibody. Consistent with findings from Fig. 2, we did not observe a statistically significant increase in MTA levels between *MTAP*-deleted and intact human GBM tumors (Supplementary Fig. 5), confirming the discrepancies between the in vitro models and primary human GBM tumors.

We further sought to test our conclusions by interrogating independent GBM metabolomic datasets in the literature, though we could not find any for which genomic information was also available. Metabolomic analyses of primary human GBM tumors are sparsely reported in the literature. The few studies that have been conducted have favored comparison between tumor populations at different stages or grades, rather than considering the unique metabolic landscape of individual tumors[33–37]. Supplementary Fig. 6a–f shows the MTA levels among 50 human GBM tumors sorted based on MTAP mRNA levels (data obtained from Supplementary Information of Prabhu et al.[38]). The frequency of homozygous *MTAP* deletion in GBM can reach 50%[39–41]. Based on an analysis of The Cancer Genome Atlas (TCGA)-GBM tumor[42,43] where both mRNA and genomic copy numbers were available, we find that, despite some exceptions, the majority of homozygous *MTAP*-deleted tumors fall in the two lower quartiles of MTAP expression (Supplementary Fig. 6g). We thus posit that, for data shown in Supplementary Fig. 6a–f, the two lower quartiles of MTAP mRNA expression would include most homozygous *MTAP*-deleted tumors, while the majority of tumors in the third and fourth quartiles would be

those that are *MTAP*-intact. The comparison of MTA levels between quartiles is statistically insignificant ($p = 0.48$, single-factor analysis of variance), which agrees with our conclusion that *MTAP*-deleted tumors do not show selective elevation of MTA.

We next interrogated another human glioma metabolomic profiling dataset from Chinnaiyan et al.[44] (Supplementary Fig. 7), where again no genomic data for individual tumors were available, which would have allowed us to determine *MTAP*-deletion status. Thus, we sought to match the metabolomic data for different glioma grades with the reported frequency of *MTAP* deletions in them. The frequency of homozygous *CDK2NA/MTAP* deletion is <3% in grade 2 glioma and <10% in grade 2/3 glioma[30], sharply contrasting the frequency of homozygous *MTAP* deletion, which is as high as 50% in grade 4 glioma (GBM)[5,39,41]. Thus, we would expect that the elevation of MTA would correspond to this deletion frequency pattern according to tumor grade, with higher MTA levels in GBM and considerably lower levels in grade 2 glioma. However, despite these different frequencies of *MTAP* deletion, MTA levels were similar in grade 2, grade 3, and grade 4 (GBM) tumors ($p = 0.15$, single-factor analysis of variance, Supplementary Fig. 7a, b). Importantly, no extreme outliers in MTA were evident. Even the highest MTA levels in individual GBM cases were about twofold higher than the median MTA levels of lower-grade glioma tumors. These data starkly contrast the extreme elevations in 2-HG levels, driven by *IDH* mutations in gliomas; these elevations were frequent in grade 2 and 3 gliomas but infrequent in GBM (Supplementary Fig. 7c, d). In addition to these MS metabolomic studies, nuclear magnetic resonance (NMR) high-resolution magic-angle spinning (HR-MAS) metabolomic profiles of GBM on a large scale ($n > 100$ cases) have not detected MTA in *MTAP*-deleted tumors even though HR-MAS can usually detect metabolites at a minimum concentration of $1 \mu M$[45]. That MTA could still not be detected in these contexts suggests that $<1 \mu M$ MTA is present in GBM tumors, which supports our conclusion that there are no glioma tumors, regardless of *MTAP* deletion status, with extreme elevations in MTA. We conclude that homozygous *MTAP*-deleted primary GBM tumors do not selectively exhibit significant elevation of MTA as opposed of what was reported in vitro, a conclusion supported by both our own results and corroborated by public domain data.

The statistically insignificant elevation of MTA (1.4-fold in absolute MTA levels, and less when correcting for total load and SAM levels) in human GBM tumors is at odds with the multiple-fold-change MTA elevations reported in vitro. Metabolomic data for MTA levels in human tumors are the measure of both intracellular and extracellular MTA. In cell culture, empirical observations indicate that differences in MTA levels between *MTAP*-deleted and WT cells are most evident in the extracellular rather than intracellular environment (Fig. 1). While we cannot distinguish intracellular and extracellular MTA in primary tumors, we can investigate the effects of *MTAP* deletion in primary tumors on intracellular PRMT5 activity inhibition. PRMT5 mediates the formation of symmetric dimethylarginine (SDMA); thus, the PRMT5 activity can be assessed by measuring the SDMA levels using the antibody against an SDMA. First, we measured the SDMA levels in *MTAP*-WT and *MTAP*-deleted cancer cells in culture (under the same conditions in which we measured MTA accumulation in media) by western blot analysis (Supplementary Fig. 8). As expected, the SDMA levels were lower in *MTAP*-deleted cells than *MTAP*-WT, indicating partial inhibition of PRMT5. Also, SDMA levels decreased following MTA treatment (50 and $100 \mu M$). These two data pieces recapitulate the literature and support the antagonistic relationship between PRMT5 activity and MTA levels. Then we measured the SDMA levels inside human GBM tumors using the same

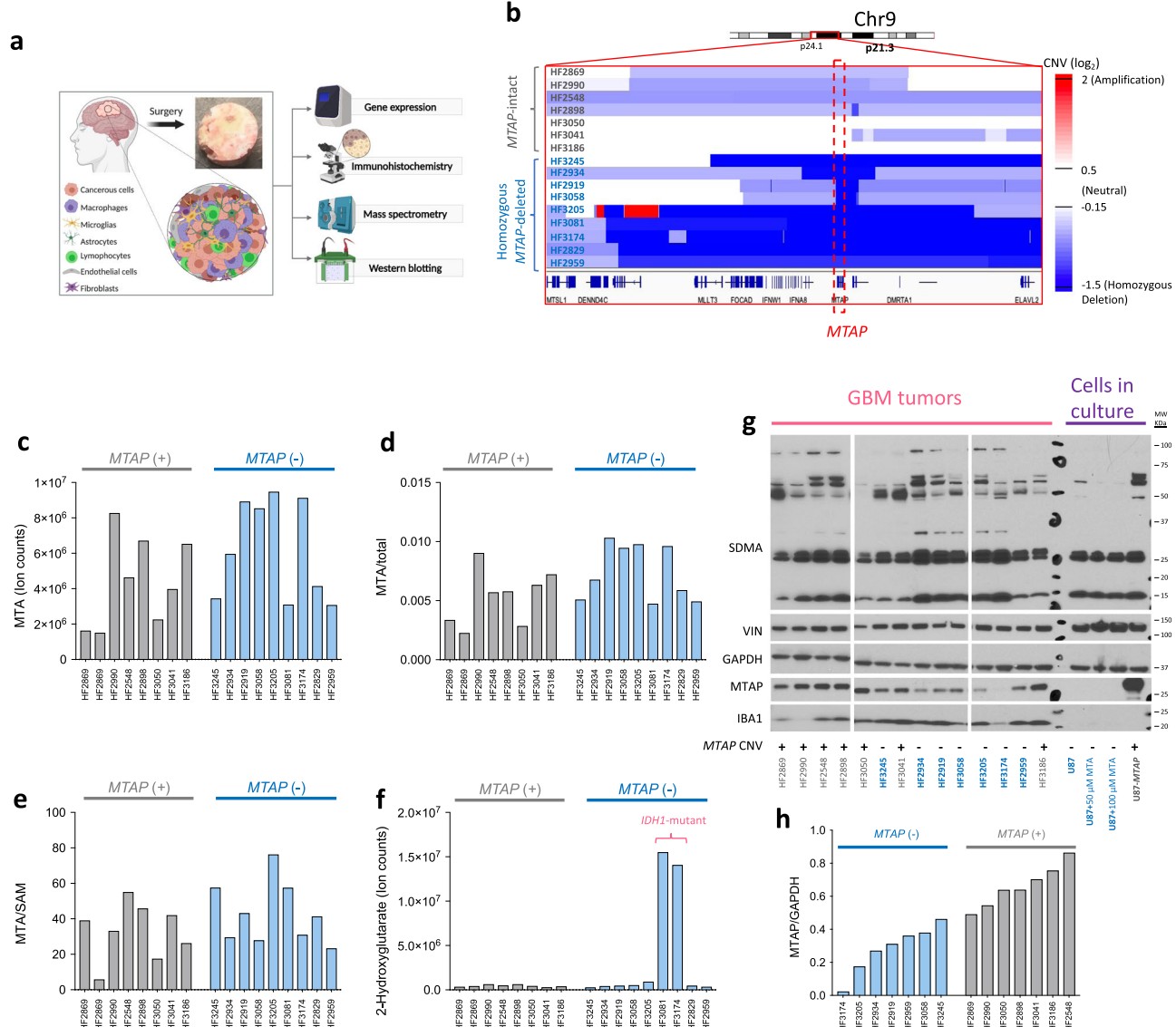

**Fig. 2 No significant elevation of MTA and no specific inhibition of PRMT5 activity in homozygous *MTAP*-deleted primary resected GBM tumors. a** GBM tumors (heterogeneous mix of transformed glioma cells and non-malignant stroma) of defined *MTAP*-deletion status (genomic profiling from Kim et al.[31]) were evaluated by mass spectrometry for MTA levels and western blot for SMDA. **b** Genomic copy-number data (dark blue: homozygous deletion, dark red: amplification) around the 9p21 locus. Each strip in the *y*-axis represents a single tumor at the specific chromosomal location (*x*-axis). **c** MTA levels in homozygous *MTAP*-deleted (*MTAP*−) versus *MTAP*-intact (*MTAP*+) GBM tumors, using the BIDMC mass spectrometric platform. Each bar represents MTA levels for each tumor ($N = 1$). There is a 1.4-fold increase in the median MTA levels (absolute ion counts) in *MTAP*-deleted tumors compared to intact tumors ($p = 0.20$, unpaired 2-tailed *t* test with unequal variance). **d**, **e** Same data but expressed normalized to total ion count for sample loading normalization (**d**) and as a ratio to SAM levels to account for methionine salvage pathway activity (**e**). Regardless of normalization, no significant elevation of MTA in *MTAP*-deleted tumors was evident. In contrast, **f** tumors with *IDH1* mutation stand out by their dramatic elevation of 2-hydroxyglutarate (2-HG), providing a positive control for genomic/metabolic correlation. **g** To evaluate PRMT5 activity, GBM tumor lysates were immunoblotted for SDMA, repeated once. As a positive control, we assessed SDMA levels in *MTAP*-deleted U87 glioma cells in culture, alone or treated with exogenous MTA versus *MTAP*-reconstituted (U87-*MTAP*) cells. SDMA levels were lower in U87 than in U87-*MTAP* cells, indicating partial inhibition of PRMT5. SDMA levels further decreased following MTA treatment. Unlike in cultured cells, no specific decrease in SDMA levels was observed in *MTAP*-deleted versus intact GBM tumors. The presence of myeloid cells in a tumor is confirmed by the myeloid marker IBA1 in tumor lysates but not seen in cells in culture. **h** MTAP protein levels corrected for loading with GAPDH control (samples derive from the same experiment and gels were processed in parallel). Due to the presence of non-malignant MTAP-expressing cells in the lysate of whole tumors, MTAP protein levels are not zero for homozygous *MTAP*-deleted tumors; however, on average, they have lower MTAP protein levels compared to intact tumors.

antibody (Fig. 2g). Unlike cell culture experiments, primary tumors showed no meaningful decrease in levels of SDMA, representing PRMT5 activity, in homozygous *MTAP*-deleted compared to *MTAP*-intact human GBM tumors. These data are consistent with the marginal elevation of MTA, intracellular or extracellular, in *MTAP*-deleted tumors. While we did not find a

correlation between SDMA levels and tumors' *MTAP* status, we noticed some variation in SDMA levels between tumors regardless of their *MTAP* status. Unlike cells in culture, homozygous *MTAP*-deleted GBM tumors express non-zero MTAP levels in western blot (Fig. 2g). This is because human GBM tumors are comprised of a mix of malignant glioma

(cancer) cells with genetic alterations like homozygous *MTAP*-deletion and non-transformed non-malignant stromal cells (e.g., myeloid cells). Thus, in homozygous *MTAP*-deleted tumors, only malignant glioma cells lack MTAP, while stromal cells express normal levels. Such non-malignant MTAP-expressing cells drive the non-zero MTAP expression (mRNA or western blot) in bulk homozygous *MTAP*-deleted GBM tumors. However, on average, the *MTAP*-deleted tumors express a lower (but non-zero) amount of MTAP protein levels by the western blot than *MTAP*-intact (Fig. 2h). The presence of myeloid cells (microglia/macrophages) in a tumor is confirmed by the myeloid marker IBA1 in a western blot of tumor lysates but not seen in cells in culture (Fig. 2g). All tumors show some levels of IBA1, verifying myeloid stromal infiltration. IBA1 levels in *MTAP*-deleted tumors nicely correlate with residual MTAP levels—i.e., the homozygous *MTAP*-deleted HF3174 tumor has the lowest IBA1 compared to other homozygous *MTAP*-deleted tumors, which also expresses the lowest but non-zero MTAP levels.

**Stromal infiltration drives the in vitro/in vivo discordance in MTA.** We next sought to identify the cause of discordance in MTA levels between cells in culture and human GBM tumors. Previous work by Sanderson et al.[11], showed that *MTAP*-deleted cells accumulate less amount of MTA when cultured in restricted methionine (3 μM) or cysteine (6 μM) RPMI than when cultured in the standard formulation with 100 μM methionine and 200 μM cysteine (data replotted in Supplementary Fig. 9). Motivated by this study, we sought to investigate the effects of culturing cells in physiological media on intracellular and extracellular levels of MTA. The composition of physiological media better represents the concentration of metabolites in human plasma[46] (Supplementary Fig. 10a). Thus, we cultured *MTAP*-deleted and WT cells in Dulbecco's Modified Eagle's medium (DMEM; historic medium) and Plasmax (physiological medium) supplemented with 2.5% fetal bovine serum (FBS) and compared the levels of MTA in the cell pellet and conditioned media. Supplementary Fig. 10b, c shows no significant difference between average MTA levels of *MTAP*-deleted cells cultured in DMEM versus Plasmax, either extracellular or intracellular. Supplementary Fig. 10d indicates that, just like in DMEM, the MTA levels are highly elevated in conditioned media of *MTAP*-deleted versus intact cells, while most modest elevations were evident in the cell pellets. These data indicate that, even in physiological media, *MTAP*-deleted cells secrete MTA. We also investigated the effects of culture medium on SDMA levels of *MTAP*-deleted and WT glioma cells (Supplementary Fig. 10f). No significant difference was observed in the SDMA levels of *MTAP*-deleted cells cultured in DMEM versus physiological media, with *MTAP*-deleted cells having lower SDMA levels. Our findings indicate that *MTAP*-deleted cells accumulate and extrude MTA when grown in a nutritional environment similar to human plasma, which would more accurately reflect the human situation. However, we do not explicitly rule out the hypothesis that nutrient differences (i.e., extremely low levels of cysteine[11]) may contribute to the discrepant observation of lack of MTA accumulation in primary human tumors with *MTAP*-deletion.

Another possible explanation could be the extrusion of MTA out of the tumor into the circulation. To investigate this possibility, we interrogated MTA levels in a metabolomic profiling dataset of cerebrospinal fluid from GBM compared and normal brain[47]. Given that the frequency of homozygous *MTAP* deletions in GBM is around 50%, we would have expected to observe a higher MTA level in the cerebrospinal fluid from at least some GBM compared to the normal brain (Supplementary Fig. 11a, b). However, the difference in MTA levels between

cerebrospinal fluid from GBM versus normal brain was nonsignificant, with no significant individual outliers. To corroborate the low likelihood of significant MTA secretion into the circulation, we compared MTA levels between venous and arterial plasma downstream and upstream of the glioma, as well as plasma obtained from a dorsal pedal vein of the same patient, control (Supplementary Fig. 11c–e; data obtained from Supplementary Information of Xiong et al.[48]). Metabolites secreted by tumors are carried downstream by venous blood, resulting in high levels of that metabolite in glioma venous compared to arterial plasma. While there were few GBM tumors in this study, given the high frequency of homozygous *MTAP* deletion in GBM, it is very likely that one or two GBM tumors to be homozygous *MTAP*-deleted in this dataset. The GBM tumor 5P had the highest MTA ratio of glioma venous to arterial. However, for the same patient, the MTA ratio of the dorsal pedal vein (control) to glioma arterial plasma was also high (Supplementary Fig. 11d), indicating high MTA production in the patient's leg. Finally, as shown in Supplementary Fig. 11e, no GBM tumor had a significantly higher ratio of MTA in the dorsal pedal vein versus glioma venous plasma than low-grade glioma. These findings indicate that MTA's secretion out of the tumor into the circulation is not a significant contributor to the discrepancy in the MTA levels between homozygous *MTAP*-deleted human GBM tumors and cells in culture.

It is already well-established that GBM tumors may have up to 75% (range 25–75%) non-malignant stromal cells, including non-transformed reactive astrocytes, microglia, macrophages, neutrophils, lymphocytes, endothelial cells, fibroblasts, and axonal and neuronal remnants[30,31,49–51]. Thus, a homozygous *MTAP*-deleted GBM tumor is an admixture of non-malignant MTAP-expressing stroma and MTAP-null malignant glioma cells. To verify the diagnosis of homozygous *MTAP* deletion and contrast this status with the extensive MTAP expression levels in non-malignant tissue, we performed immunohistochemistry on intracranial xenografted gliomas in mice. We first validated an MTAP rabbit monoclonal antibody by demonstrating intense staining in formalin-fixed paraffin-embedded (FFPE) sections of *MTAP*-WT tumors and loss of staining in *MTAP*-deleted tumors generated from known *MTAP*-genotype cell lines (Supplementary Figs. 12–14). Every xenograft established from *MTAP*-WT (or isogenic *MTAP*-rescued) cancer cell lines showed intense staining, while all xenografts established from *MTAP*-deleted cell lines showed no staining. Simultaneously, non-malignant stromal cells such as microglia, lymphocytes, and endothelial cells stained positive for MTAP. Together, these staining results support that this antibody accurately quantifies MTAP levels in FFPE slides. Notable among our stained slides was the significant amount of positive staining in the normal mouse brain and stromal cells, especially given that xenografts typically grow much less invasively and are less populated by stromal cells compared to human GBMs (Supplementary Fig. 15). By extension, true human GBMs are more likely to contain a higher proportion of stromal components such as astrocytes and myeloid cells.

Next, we applied this antibody to FFPE primary GBM sections. Of the 60 cases analyzed, we found that approximately 40% of cases showed a complete lack of MTAP staining in cancer cells, which generally corresponds to the expected *MTAP*-deleted frequency in GBM[41]; notably, intense staining was observed in non-malignant stromal cells. *MTAP*-intact cases showed intense, uniform staining in both glioma and stromal cells (Fig. 3, Supplementary Fig. 17). The histology of *MTAP*-deleted primary GBM cases appeared similar to the *MTAP*-deleted intracranial xenografted cases, except that the degree of *MTAP*-positive stromal infiltration was much more significant in the primary human GBM tumors compared to xenografts (Supplementary

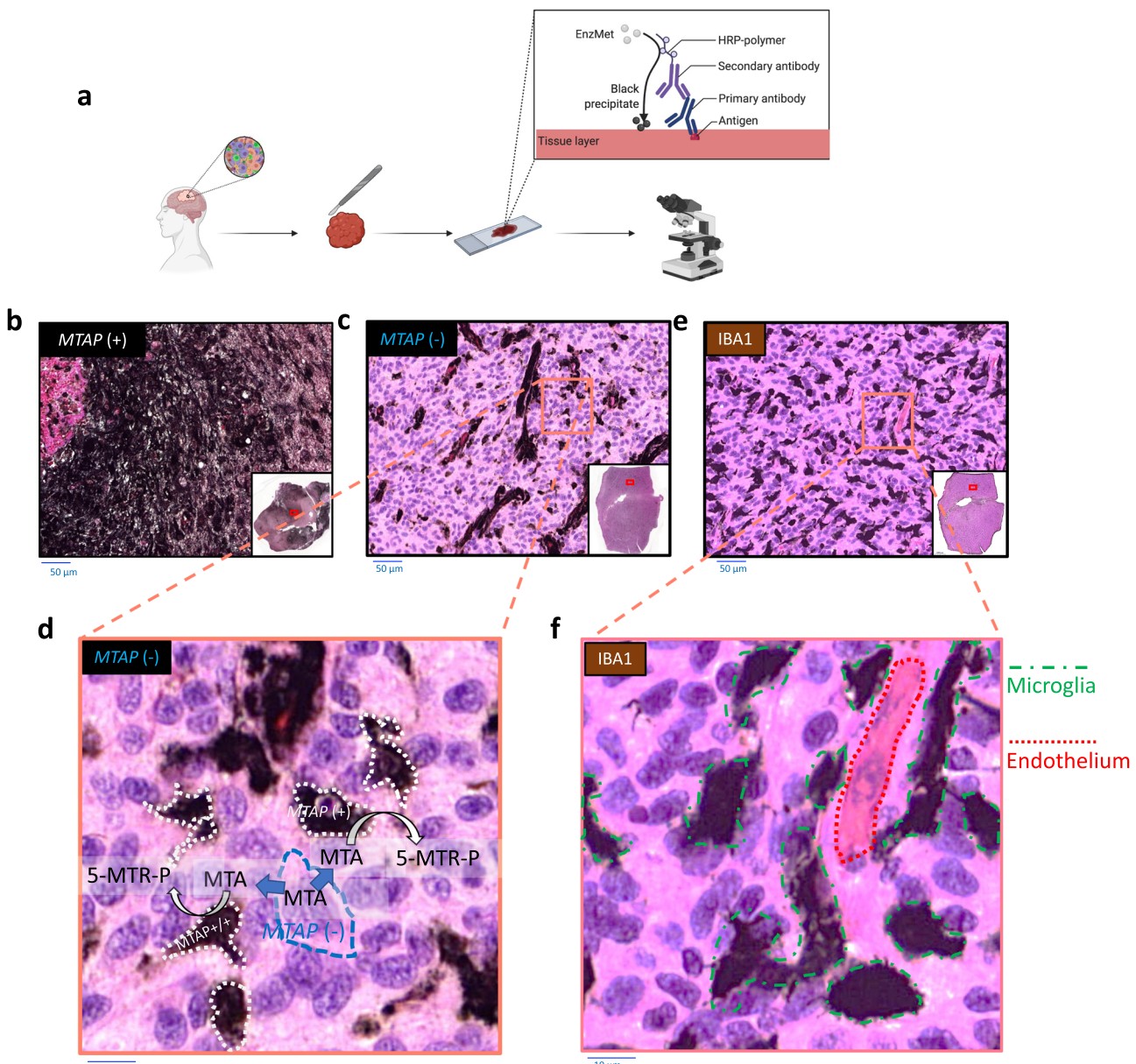

**Fig. 3 Primary GBM tumors are extensively infiltrated by MTAP-expressing non-malignant stromal cells. a** Immunohistochemistry (IHC) for MTAP and IBA1 on formalin-fixed paraffin-embedded (FFPE) sections of GBM tumors. **b**, **c** IHC was performed with monoclonal anti-MTAP. The slides were developed with EnzMet (black staining indicates MTAP expression) and counterstained with hematoxylin (blue, nuclei) and eosin (pink, cytosol, and extracellular space) on at least 50 cases stained with MTAP and IBA1 and representative of three independent cases are shown. Representative sections of *MTAP*-positive (**b**) and an *MTAP*-negative GBM tumor (**c**); x20 objective. Note the extensive MTAP-positive staining areas on the background of otherwise non-staining *MTAP*-negative glioma cells; histopathologic evaluation of morphology indicates that these *MTAP*-positive cells correspond to stromal components, including microglia, endothelial cells, and activated astrocytes. **d** Higher magnification view of **c** golden square. Any MTA secreted by *MTAP*-deleted glioma cells (blue dashed outline) stands to be phosphorylated by MTAP (producing 5-MTR-P) either released or taken up by MTAP-expressing stromal cells (white dashed outlines). **e** A representative section of an *MTAP*-negative GBM tumor stained against IBA1 antibody (myeloid marker), indicating the extensive presence of myeloid cells (microglia/macrophages) inside GBM tumors. **f** Higher-magnification view of **e**.

Fig. 15). Supplementary Fig. 16a–c shows the strong correlation between the residual MTAP staining and myeloid content in *MTAP*-deleted GBM tumors. In homozygous *MTAP*-deleted tumors, the residual MTAP staining is restricted to stromal cells (no MTAP staining in malignant glioma cells), predominantly of myeloid origin (IBA1-positive). We also interrogate an independent dataset (TCGA[41]) for the correlation between mRNA levels of MTAP and AIF1 (the official gene symbol of IBA1, microglia/macrophages marker) among GBM tumors with known *MTAP* status. Supplementary Fig. 16d indicates a significant positive correlation between mRNA levels of MTAP and AIF1 in *MTAP*-deleted tumors versus intact ones. However, no positive correlation was observed between MTAP and AIF in *MTAP*-intact tumors since both glioma and stromal cells express MTAP in *MTAP*-intact tumors. This figure also indicates that homozygous *MTAP*-deleted GBM tumors, on average, have lower but non-zero levels of MTAP mRNA levels compared to *MTAP*-intact tumors, supporting our western blot data in Fig. 2g, h and validate that IBA1-positive myeloid cells drive the non-zero MTAP expression in bulk *MTAP*-deleted GBM tumors.

Having identified *MTAP*-WT cells' presence (i.e., stromal cells, which are predominantly myeloid in GBM) inside *MTAP*-deleted tumors, we hypothesized that excess MTA secreted from homozygous *MTAP*-deleted cancer cells may be metabolized by non-malignant MTAP-expressing cells present inside tumors. To test this hypothesis, we cultured *MTAP*-WT macrophages with exogenous MTA and measured MTA levels in conditioned media using proton NMR ($^1$H-NMR). In the Supplementary Fig. 18a, b, no MTA peaks were detected from the conditioned media of *MTAP*-WT macrophages (RAW-264.7 and MV-4-11) incubated with exogenous MTA compared to media from a control plate (DMEM) or conditioned media from *MTAP*-deleted leukemia cells (CCRF-CEM) under the same experimental condition. This result indicates that either *MTAP*-WT cells release functional MTAP enzyme into the media or take up MTA from the media and metabolize it. First, we sought to see whether the release of functional MTAP enzyme by *MTAP*-WT cells is responsible for eliminating exogenous MTA from conditioned media. We grew *MTAP*-WT RAW-264.7 (macrophages) cells in DMEM; after 3 days, we collected the conditioned media and spun it down to remove any cells, then we incubated the conditioned media collected from RAW-264.7 cells with 20 μM exogenous MTA for 3 more days. Supplementary Fig. 19a shows that exogenous MTA was not eliminated from the cell-free conditioned media from macrophages. This result indicates that the release of functional MTAP enzymes does not contribute to the disappearance of exogenous MTA from the conditioned media of *MTAP*-WT cells. Next, we sought to investigate whether exogenous MTA is taken up and metabolized by *MTAP*-WT cells. Thus, we cultured *MTAP*-WT macrophages (RAW-264.7) or glioma cells (D423) with labeled methyl tri-deuterated-MTA (D3-MTA) (Fig. 4a). We observed rapid elimination of labeled D3-MTA from the conditioned media of *MTAP*-WT cells followed by deuterium enrichment of intracellular MTA, methionine, and SAM (Fig. 4b, c, and Supplementary Fig. 19c, d). Figure 4d shows the deuterium's fate from the labeled D3-MTA into methionine (methionine salvage pathway) and SAM (polyamine biosynthesis). This result indicates that extracellular MTA can be taken up and metabolized by *MTAP*-WT cells.

Next, we co-cultured *MTAP*-deleted glioma cells with macrophages (RAW-264.7), immortalized normal human astrocytes, and isogenic *MTAP*-rescued cells (U87-*MTAP*) and measured MTA levels in conditioned media using $^1$H-NMR. We observed robust accumulation of MTA in the conditioned media of *MTAP*-deleted cells but not of *MTAP*-WT or *MTAP*-rescued cells (Fig. 4e). When *MTAP*-deleted U87, SW1080, HT1080, and SKMEL5 cells were co-cultured with *MTAP*-WT macrophages (Fig. 4e) or normal human astrocytes (Supplementary Fig. 18c), no MTA peaks were detected in the conditioned media by $^1$H-NMR. Furthermore, the co-culture of U87 parental *MTAP*-deleted glioma cells with U87 *MTAP*-rescued cells resulted in the disappearance of MTA peaks in the $^1$H-NMR spectrum of the conditioned media (Supplementary Fig. 20). These results indicate that secreted MTA from *MTAP*-deleted cells is consumed (metabolized) by *MTAP*-WT cells; thus, co-culture of *MTAP*-deleted with *MTAP*-WT cell lines abrogates MTA accumulation in conditioned media. We also confirmed secretion and consumption of MTA when *MTAP*-deleted cells are co-cultured with macrophages in physiological media (Plasmax, Supplementary Fig. 18d). Our findings indicate that *MTAP*-WT cells in the vicinity of *MTAP*-deleted cancer cells prevent extracellular accumulation of MTA.

We also investigated the effects of co-culturing *MTAP*-deleted and *MTAP*-intact cells on PRMT5 activity—intracellular MTA levels—of *MTAP*-deleted cells. We earlier discussed the discrepancies in the magnitudes of reported MTA between *MTAP*-deleted and *MTAP*-WT cancer cell lines. To avoid such problems, we used a different approach to assess MTA levels inside cells indirectly. Since intracellular MTA inhibits PRMT5 activity and subsequently decreases SDMA levels, assessing SDMA levels by western blot is an indirect measurement of intracellular MTA. *MTAP*-deleted and *MTAP*-rescued cells were co-cultured with suspended leukemia MV-4-11 (monomyelocytic leukemia, macrophage-like) cells in DMEM. After 3 days, suspended leukemia cells were removed, and adherent cells were washed twice with phosphate-buffered saline (PBS) before they were lysed. On western blot (Fig. 4f), monoculture *MTAP*-deleted cells had lower SDMA levels, indicating lower PRMT5 activity and higher intracellular MTA levels than co-cultured *MTAP*-deleted cells. These findings emphasize that the co-culture of *MTAP*-deleted cells with *MTAP*-intact cells negates the selective vulnerability caused by *MTAP* deletion, thereby indicating that *MTAP*-deletion sensitivity in a heterogeneous human GBM tumor may be attenuated by the presence of stroma in the tumor microenvironment.

## Discussion

We undertook this study in response to the emergence of homozygous *MTAP*-deleted precision therapy endeavors despite a lack of exploration of the metabolic state of primary tumors. As the therapeutic window of therapies directed at *MTAP* deletion relies on massive intracellular accumulation of MTA[9,10,15,52–54], ensuring that this in vitro phenotype is consistent in human tumors was a natural and urgent area for investigation. Perhaps surprisingly, none of the recent papers documenting *MTAP*-dependent vulnerabilities reported MTA measurements in primary human tumors or even xenografted tumors of defined *MTAP*-deletion status, despite the significant efforts that have been built on the supposition that observed elevations in MTA in vitro would be reflected in primary tumors[9,10,15,52–54].

We took to the literature—keen on finding studies describing MTA levels in distinct, genetically defined homozygous *MTAP*-deleted primary tumors—but with little success. We could not find any literature reports directly measuring MTA levels in genomically identified *MTAP*-deleted than *MTAP*-intact primary human tumors, whether GBM or others. This was quite surprising, given that MTA is a metabolite frequently detected in many metabolomic profiling studies.

To fill this gap, we interrogated our metabolomic datasets generated from different sets of GBM tumors using two independent metabolic platforms (BIDMC and Metabolon, Inc.). Our data suggest a statistically insignificant increase in MTA levels between homozygous *MTAP*-deleted and *MTAP*-intact tumors. This marginal elevation in MTA levels was at odds with the vast elevations in MTA reported for *MTAP*-deleted cell lines in vitro[9,10,15]. To investigate whether the in vitro metabolic vulnerability posed by *MTAP* deletion holds in human GBM tumors, we assessed PRMT5 activity by measuring SDMA levels in the same set of human tumors in which we performed metabolomics. Unlike in cell culture experiments, we did not observe lower SDMA levels in homozygous *MTAP*-deleted tumors compared to intact ones. As an independent verification of our findings, we queried the public domain datasets of metabolomic studies performed on human GBM tumors[33,44]. We did not observe statistically significant MTA elevation in GBM tumors with low MTAP expression compared to those with high MTAP expression or GBM tumors with frequent *MTAP* deletions and grade 2 and 3 glioma tumors with rare *MTAP* deletions. If *MTAP* deletions do indeed lead to MTA accumulation in primary human tumors, we would expect to observe dramatically higher MTA levels (on the order of >20-fold, as seen in in vitro intracellular data from

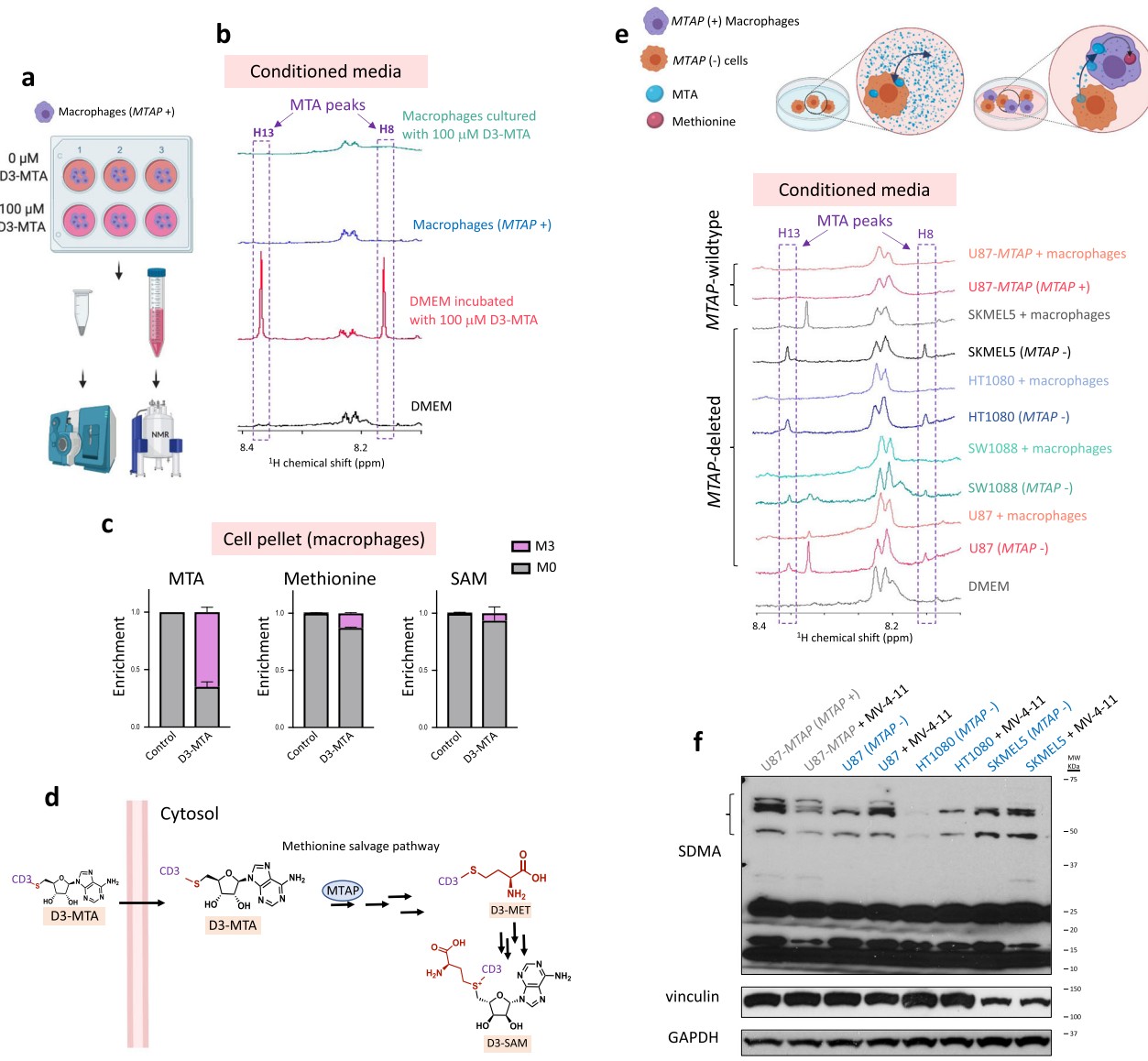

**Fig. 4 Exogenous MTA is consumed by *MTAP*-intact cells through the methionine salvage pathway. a** *MTAP*-WT macrophages (RAW-264.7) were cultured with 100 μM methyl tri-deuterated-MTA (D3-MTA). Cells and conditioned media were extracted and prepared for LC-MS and NMR measurements, respectively. **b** Exogenous D3-MTA rapidly disappears from the media in cultures of macrophages, as the deuterium label (Mass + 3; M + 3) appears in intracellular methionine and SAM (**c**), consistent with *MTAP*-dependent metabolism by the methionine salvage pathway (**d**). Data in panel **c** are expressed as mean + SD, N = 3 biological replicates. **e** *MTAP*-deleted cancer cells accumulate MTA in conditioned media (detected by NMR) while secreted MTA is consumed by *MTAP*-intact macrophages, abrogating MTA accumulation in the extracellular environment. **f** Co-culture of *MTAP*-deleted cancer cell lines with *MTAP*-intact myeloid cells prevents loss of SDMA, indicating maintained PRMT5 activity, repeated once. *MTAP*-deleted and reconstituted glioma cells were co-cultured with myeloid leukemia cells (MV-4-11, a monomyelocytic suspension cell line). Before lysate preparation, suspended MV-4-11 myeloid cells were removed, and cancer cells were extensively washed. For each independent *MTAP*-deleted cell line, SDMA levels increased with myeloid co-culture, indicating restoration of PRMT5 activity.

Fig. 3c in that publication[9]) in a substantial fraction of GBM tumors. Our study evaluated MTA in bulk GBM tumors, which are a heterogeneous mixture of cancer and stromal cells. One shortcoming of performing metabolomics and western blots on such tumors is that the measured signal (i.e., MTA levels and SDMA levels) derived from both cancer and stromal cells. Although signal derived from cancer cells is diluted by stromal cells, significant metabolic aberrations such as elevation of 2-HG can still be seen in resected bulk tumors (Fig. 2f and Supplementary Fig. 7). However, our data of profiling GBM tumors from two different dataset (HF series and MDA series) demonstrate that there are nonsignificant 1.4-fold (p = 0.2) and 1.14-fold (p = 0.9) increase in MTA levels in *MTAP*-deleted tumors versus intact ones. These numbers are too low to be explained by signal dilution caused by stromal contamination if cell culture data are taken as a reference point. However, our data do not rule the possibility of minor accumulation of MTA in *MTAP*-deleted glioma cells; just that such accumulation would have to be much less than reported in culture.

In sum, the extreme elevation of MTA in homozygous *MTAP*-deleted cells in vitro did not extend to our metabolome analyses of primary GBM tumors. A variety of reasons may explain this discrepancy between cell culture and human tumor data. The explanation we favor centers on the observation that *MTAP*-deleted cells excrete MTA into the media (Fig. 1h). In the literature, we noticed large discrepancies between different studies

in the magnitudes of reported MTA increases in *MTAP*-deleted versus *MTAP*-intact cancer cell lines (Supplementary Fig. 2). Careful analysis of these results in light of our own in vitro data points to MTA secretion out of homozygous *MTAP*-deleted cells. Thus, the discrepancies between different studies appear to arise from how well intracellular and extracellular metabolite levels are differentiated during sample processing. Secretion of MTA by *MTAP*-deleted cells in agreement with previous studies[19,55] further corroborate this finding.

An additional explanation suggested by our findings is that human GBM tumor environments are extensively infiltrated by MTAP-expressing non-malignant stromal cells (Fig. 3 and Supplementary Figs. 15 and 16). To investigate the effects of *MTAP*-WT cells in the vicinity of *MTAP*-deleted cells on intracellular and extracellular MTA levels, we co-cultured *MTAP*-deleted cells with *MTAP*-WT ones. We observed that this co-culture abrogated the accumulation of MTA in the conditioned media of *MTAP*-deleted cells. Furthermore, we found that the co-culture of *MTAP*-deleted cells with suspended *MTAP*-WT leukemia cells (MV-4-11, myeloid-like) resulted in higher SDMA levels—indicating higher PRMT5 activity and lower intracellular MTA levels—compared to the monoculture *MTAP*-deleted cells. Also, culturing *MTAP*-WT cells with labeled D3-MTA revealed labeling of intracellular MTA, methionine, and SAM, evidencing MTA uptake, consumption, and metabolism by *MTAP*-WT cells. These findings suggest that the metabolism of MTA by stromal cells is likely the predominant cause of the absence of any detectable increase in MTA in primary GBMs. However, our study does not rule out other factors that could influence MTA accumulation in *MTAP*-deleted cells, i.e., extreme cysteine deficiency shown by Sanderson et al.[11] (though physiological media Plasmax still yielded MTA secretion in *MTAP*-deleted glioma cells, Supplementary Fig. 10). Our findings highlight the metabolic discrepancies between in vitro models and primary human tumors. It is already well understood that cell culture conditions do not always accurately reflect physiologic, metabolic conditions[11,56,57]: our study exemplifies another aspect, stromal infiltration, as a complicating factor for tumor metabolism. Our results suggest more systematic efforts to validate cell culture and xenograft studies on cancer metabolism with primary human tumor data.

That *MTAP* deletions can be leveraged as a point of selective vulnerability in various cancers has spurred much excitement. Our analysis challenges whether the metabolic conditions required for therapies to exploit vulnerabilities associated with elevated MTA/*MTAP* deletions are present in primary human tumors, giving pause to whether these would translate to the clinic. However, our findings do not rule out that MAT2A and/or PRMT5 inhibitors could prove useful in oncology. Merely, such inhibitors would have to achieve a therapeutic window through a mechanism other than MTA accumulation. Indeed, recent research has already pointed toward *MTAP*-deletion-independent sensitization mechanism to PRMT5 inhibition[58]. Some GBM tumors in our dataset showed relatively low SDMA levels—suggesting low PRMT5 activity—without *MTAP* deletions (Fig. 2g). Perhaps such tumors with low PRMT5 activity could be suitable candidates for targeted therapies against PRMT5 or MAT2A, provided a reliable molecular marker can be found to identify them.

## Methods

### Identification of cell lines with *MTAP*-homozygous deletions. 
Scoring of homozygous deletions, even with robust copy-number data, can be problematic owing to uneven ploidy across cell lines, as well as where deletion boundaries only partially cover the gene yet eliminate its functional expression. Thus, we sought to identify functionally *MTAP*-null cell lines. For the NCI-60 cell line comparisons, we gathered data from the Sanger Center Catalogue of Somatic Mutations in Cancer[59]. We used confirmed data from cBioPortal with the NCI-60 panel

data[43,60] and DepMap from the Cancer Cell Line Encyclopedia and the Genentech collection[61]. Where deletion calls were ambiguous (e.g., SF-295), we looked at mRNA expression data or literature western blot data[9,10,15,52] to decide what constituted a functionally *MTAP*-null cell line. In the NCI-60 cell line panel, the following cell lines lack *MTAP*: NCI-H322, SKMEL5, K562, SF268, MALME3M, ACHN, MDAMB231, OVCAR5, CCRF-CEM, SR, MCF7, and A549.

### Cell culture and xenograft generation. 
The cell lines used in this study that are *MTAP*-WT were D423 (CVCL_1160, H423/D423-MG) and D502 (CVCL_1162, H502[62]), which were kindly provided by Darrel Bigner[62]. U343 (CVCL_S471, U343-MG[63]), LN319 (CVCL_3958, a sub-clone of LN-992[64]), and NB1 (CVCL_1440) were obtained from the Department of Genomic Medicine/Institute for Applied Cancer Science Cell Bank at MD Anderson. MV-4-11 (CVCL_0064) was purchased from NCI, RAW-264.7 (CVCL_0493) was purchased from ATCC (TIB-71), and immortalized normal human astrocytes was kindly provided by Dr. Seth Gammon (Cancer System Imaging). The *MTAP*-deleted cell lines U87 (CVCL_0022), SW1088 (CVCL_1715), and SKMEL5 (CVCL_0527) were obtained from the Department of Genomic Medicine/Institute for Applied Cancer Science Cell Bank at MD Anderson. Gli56 (D. Louis) and CCRF-CEM (CVCL_0207) were purchased from NCI. HT1080 (CVCL_0317) was kindly provided by Dr. Seth Gammon (Cancer Systems Imaging). All cell culture experiments were conducted according to the provider's instructions and as previously described[65]. The U87 *MTAP*-rescued cell line was generated using the same procedure we previously used to generate an *ENO1*-rescued cell line[1]. The MTAP cDNA was obtained from Life Technologies, sequence-verified, and cloned into a pCMV GFP lentiviral vector using Gateway cloning technology. 293T cells were transfected with viral plasmids using polyethyleneimine. The viral supernatant was collected 72 h after transfection and added to the U87 cell line for infection.

All cells were cultured at 37 °C in a 5% $CO_2$ atmosphere at pH 7.4 in ATCC-suggested media (DMEM) unless stated otherwise. DMEM has 4.5 g/l glucose, pyruvate and glutamine (Cellgro/Corning, #10-013-CV) with 10% (20% for Gli56) fetal bovine serum (Gibco/Life Technologies, #16140-071), 1% penicillin–streptomycin (Gibco/Life Technologies, #15140-122), and 0.1% amphotericin B (Gibco/Life Technologies, #15290-018). All cell lines were confirmed as mycoplasma negative by enzyme-linked immunosorbent assay using the MycoAlert PLUS Detection Kit (Lonza) and were authenticated by short tandem repeat DNA fingerprinting and chromosomal analysis by the Cytogenetics and Cell Authentication Core at MD Anderson.

MD Andersons's Institutional Animal Care and Use Committee approved all procedures for animal studies. Xenografts with the D423 (*MTAP*-WT), U87 (*MTAP*-deleted), U87 pCMV MTAP (*MTAP*-rescued), NB1 (*MTAP*-WT), and SKMEL5 (*MTAP*-deleted), and Gli56 (*MTAP*-deleted) cancer cell lines were generated as detailed below for FFPE sections. For subcutaneous xenografts, 5 million cells were injected in the flanks of a nude athymic mouse (bred by MD Anderson's Department of Experimental Radiation Oncology). Mice were housed in a room with an ambient temperature between 20 and 22 °C and humidity of 30–70%, in a 12-h light–12-h dark cycle. Tumors were harvested and fixed in 4% phosphate-buffered formalin. Dehydration, paraffin embedding, and tissue sectioning were performed by MD Anderson's Veterinary Pathology Core. Intracranial glioma cell injections were performed by the MD Anderson Intracranial Injection Core at MD Anderson (Dr. Fred Lang, Director[66]). Immunocompromised female nude Foxn1nu/nu mice were first bolted. After 2 weeks, the mice were injected with 200,000 cells into the brain through the bolt by the MD Anderson Intracranial Injection Fee-for-Service Core[66]. Formalin-fixed brains with xenografted glioma tumors were submitted to the Veterinary Pathology Core for embedding and sectioning.

### Metabolomic profiling of polar metabolites. 
Polar metabolites were profiled using the BIDMC platform. The extraction of samples was performed in the house as follows for each specific set of samples. Polar metabolites of GBM tumors were also profiled through fee for service by Metabolon, Inc. (Durham, NC).

### Targeted MS. 
A hybrid QTRAP 5500 triple quadrupole mass spectrometer (AB SCIEX), which is coupled to a Prominence UFLC HPLC system (Shimadzu), was used. First, each sample was resuspended in 20 μl high-performance liquid chromatography (HPLC)-grade water, then 5–7 μl of it was injected into the spectrometer. The selected reaction monitoring (SRM) method was used for steady-state analyses with 262 endogenous water-soluble metabolites with a dwell time of 3 ms for each SRM, and a total cycle time of 1.55 s, with 10–14 data points acquired for each metabolite. Voltages used for the positive and negative ion modes were +4950 and −4500 V, respectively. Hydrophilic interaction chromatography using a 4.6 mm × 10 cm Amide XBridge column (Waters) at 400 μl/min was used to deliver samples to the spectrometer. Following gradient sequence were used during the measurement: 85% buffer B (HPLC grade acetonitrile) to 42% B from 0 to 5 min, followed by 42% B to 0% B from 5 to 16 min, with no gradient from 16 to 24 min, and finally, gradients were run from 0% B to 85% B from 24 to 25 min, and 85% B was held for 7 min to re-equilibrate the column. Buffer A comprised 20 mM ammonium hydroxide/20 mM ammonium acetate (pH 9.0) in 95:5 water:

acetonitrile. MultiQuant v2.1 software (AB SCIEX) was used to integrate total ion current peak.

**Intracellular metabolites from cells in culture (cell pellet)**. Adherent cancer cells growing in 10-cm plates at 50–90% confluency were harvested for metabolomic profiling as follows. First, media was removed (and extracted for its profiling), and the plated cells were washed twice with ice-cold 1× PBS (Corning). Cells were then placed in dry ice, and we added 4 ml of 80% methanol pre-cooled to −80 °C. Then they were incubated in the −80 °C freezer for 20 min. Cells were scraped off the plate while on dry ice, placed in pre-cooled tubes, and finally centrifuged at maximum speed, $17,000 \times g$ for 5 min at 4 °C. The supernatant containing polar metabolites was concentrated using an Eppendorf Vacufuge Plus[67] and submitted to the BIDMC MS core.

**Secreted metabolites (conditioned media)**. A total of 1–2 ml of conditioned media was centrifuged at maximum speed for 10 min to remove any unblocked cells or cell debris. Subsequently, 4 volumes of −80 °C pre-cooled methanol for every 1 volume of media was added to make the final concentration of 80% (vol/vol) methanol solution, then vortexed and left at −80 °C for 6–8 h to precipitate serum and extract polar metabolites. Ice-cold methanol media mix was centrifuged at maximum speed ($17,000 \times g$) for 10 min at 4 °C. Following centrifugation, the supernatant was separated and dried in the vacuum concentrator (Eppendorf Vacufuge Plus) and then sent to the BIDMC core[67].

**Frozen tumors**. GBM tumors were collected under the protocol approved by the institutional review board at MD Anderson Cancer Center (PA15-0940, LAB03-0320, and LAB03-0221). Under the protocol, informed written consent from patients were obtained. Tumors snap-frozen in liquid nitrogen were kept at −80 °C until extraction and never thawed. Every effort was taken to keep tumors frozen until they were placed in cold methanol. Frozen tumors were weighed on dry ice without thawing, and we placed them in dry ice pre-cooled Fisher Tube (#02-681-291) with Qiagen steel beads. Then we added 1 ml of 80% at −80 °C pre-cooled methanol to each tube. Then we shook the tubes with Qiagen TissueLyser for 45 s at room temperature at 28 Hz for multiple rounds. We placed the tubes in the dry ice between each round. After samples became homogenous, the samples' final volume was proportioned to 50 mg of tissue/2 ml of 80% methanol. After this step, we kept the samples at −80 °C for 24 h and then vortexed each sample and centrifuged them at maximum speed ($17,000 \times g$) for 10 min at 4 °C. Then we collected the supernatant and stored it at −80 °C until vacuum concentration. For each study, we used Eppendorf Vacufuge Plus to vacuum-concentrate the same volume of samples. Dried samples were sent to the BIDMC core.

**Determination of MTA by ¹H-NMR in conditioned cell culture media**. Conditioned media from cells was collected and centrifuged for 10 min to remove cells and cell debris. Post-centrifugation media (2 ml) was transferred to a new tube, where 4 (8 ml) volumes of methanol were added to precipitate proteins. The sample was then vortexed and centrifuged at the highest speed for 10 min, and the supernatant was transferred to a new tube. The post-centrifuged supernatant (8 ml) was then vacuum concentrated using the Eppendorf Vacufuge Plus followed by lyophilization for 24 h; the resulting pellet was resuspended in 600 μl of D-DMSO (dimethyl sulfoxide-d6; Alfa Aesar, 99.5%, #A16893-18). Any insoluble material was removed by centrifugation. The clear D-DMSO solution was then placed in a 5-mm NMR tube for NMR analysis. ¹H-NMR measurements were performed on a Bruker Avance III HD 500 MHz equipped with a CryoProbe Broadband Observe probe or a Bruker Avance One 600 MHz equipped with a 5-mm TXI probe at MD Anderson Cancer Center. The resulting spectrum was obtained using the zg30 pulse sequence with either 100 or 1024 scans and a relaxation delay equal to 1 s. Spectrum analysis was performed using the Bruker TopSpin 3.1 software. For MTA, the adenosine protons (HMDB0001173), with chemical shifts of 8.15 and 8.35 ppm, were readily detectable in the supernatant of *MTAP*-deleted media extract, but not *MTAP*-intact or *MTAP*-rescued extract, and in the supernatant of co-cultured *MTAP*-deleted and *MTAP*-intact media extract. The most intense MTA peak occurred at 2.1 ppm, corresponding to the methyl group protons adjacent to the sulfur. However, in the media extract of *MTAP*-deleted cells, this peak was obscured by more abundant metabolites. The peaks for the ribose group were of low intensity and not detectable. We validated our observed chemical shifts by spiking sample extracts with pure MTA standard (Sigma-Aldrich, #D5011, 100 mg).

**Absolute quantification of MTA in the cell pellet and the conditioned media**. *MTAP*-deleted and WT cells were grown in DMEM or Plasmax supplemented with 2.5% FBS. After 3 days, the cell pellet and the conditioned media were harvested for quantification of MTA. Media was collected, spun down to remove any cells, and extracted with 80% methanol. Cells were washed with ice-cold 1× PBS (Corning) and placed in dry ice. Two hundred microliters of the extraction buffer (40% methanol: 40% acetonitrile: 20% water) pre-cooled to −20 °C. Cells were scraped off the plate while on dry ice placed in pre-cooled tubes. The concentration of MTA in the extract of cell pellet and extract of media was determined in LC-MS. A quadrupole-orbitrap mass spectrometer (Q Exactive, Thermo Fisher Scientific, San Jose, CA) coupled to hydrophilic interaction chromatography was used for MTA quantification. LC separation was on an XBridge BEH Amide column (2.1 mm × 150mm × 2.5 mm particle size, Waters, Milford, MA) using a gradient of solvent A (20 mM ammonia acetate, 20 mM ammonium hydroxide in 95:5 water: acetonitrile, pH 9.45) and solvent B (acetonitrile). The flow rate was 150 μl/min. LC gradient was: 0 min, 85% B; 2 min, 85% B; 3 min 80% B; 5 min, 80% B; 6 min, 75% B; 7 min, 75% B; 8 min, 70% B; 9 min, 70% B; 10 min, 50% B; 12 min 50% B; 13 min, 25% B; 16 min, 25% B; 18 min, 0% B; 23 min, 0% B; 24 min, 85% B. MS scans were performed from $m/z$ 70–1000 at 140,000 resolution. For MTA quantification, deuterium-labeled MTA was used as an internal standard. Data were analyzed using the EI-Maven 7.0 software (Elucidata, LLC., elucidata.io.), and isotope labeling was corrected for the natural abundance. The absolute amount of MTA in the total cell pellet and media (cell pellet approximately 3 M cells, media 3 ml) was calculated from this concentration. The cell density was determined by harvesting cells with trypsin and counting cells using trypan blue.

**Tracing deuterated MTA**. *MTAP*-WT mouse macrophages (RAW-267.4) and glioma cells (D423) were cultured in a 6-well plate with and without 100 μM deuterated MTA in DMEM for 1 day. Then cell culture media were collected and prepared for NMR studies. Cells were washed with ice-cold 1× PBS (Corning) and placed in dry ice. We then added 200 mM of the extraction buffer (40% methanol: 40% acetonitrile: 20% water) pre-cooled to −20 °C. Cells were scraped off the plate while on dry ice placed in pre-cooled tubes and finally centrifuged.

**Validation of an anti-MTAP rabbit monoclonal antibody for the immunohistochemistry**. To confirm homozygous deletions *MTAP* by immunohistochemistry in primary tumor FFPE sections, we first validated a monoclonal antibody for this application by demonstrating staining in FFPE slides of xenografted tumors with a known MTAP genotype. Xenografts generated with the D423 (*MTAP*-WT), U87 (*MTAP*-deleted), U87 pCMV MTAP (*MTAP*-rescued) NB1 (*MTAP*-WT), SKMEL5 (*MTAP*-deleted), and Gli56 (*MTAP*-deleted) cancer cell lines were fixed in formalin and embedded in paraffin. Immunohistochemistry was performed on coronal sections of mouse brain xenografted with human glioblastoma cells. Tissue was fixed in formaldehyde before being embedded in paraffin. The tissue was sliced to the desired thickness and embedded onto the slide. Slides were left to incubate at 60 °C overnight. To detect MTAP, sections were subjected to antigen retrieval in citrate buffer (1:100 Vector Laboratories Antigen Unmasking Solution [Citrate-Based], H-3300, 250 ml) at high pressure for 10 min. Sections were covered with a blocking buffer of 2% goat serum (Vector Laboratories, S-1000, Normal Goat Serum, 20 ml) in PBS (Quality Biological PBS 10×, pH 7.4, 1000 ml) for 1 h. Sections were then covered with anti-MTAP rabbit monoclonal (Abcam, #ab126623, clone EPR6892, Lot: GR97816-5, Lot: YI070108C5) in a 1:200 dilution with 2% goat serum in PBS and left to incubate overnight at 4 °C. Sections were then washed in PBS and incubated for 30 min with 1× goat anti-rabbit IgG secondary antibody, poly-horseradish peroxidase conjugate (Invitrogen by Thermo Scientific, Ref: B40962, Lot: 2140280). After being washed in PBS and Tween 20 (Fisher BioReagents, #BP337-500), sections were developed using either Impact NOVAred (Vector Laboratories; yields a red to brown color for stain) or using EnzMet (Nanoprobes, #6001, 30 ml; yields a black stain). For NOVAred, slides were then counterstained using hematoxylin; for EnzMet, slides were counterstained with hematoxylin and, optionally, an eosin counterstain. Sections were mounted using Denville Ultra Microscope Cover Glass (#M1100-02) and Thermo Scientific Cytoseal 60 and left to dry overnight. There was an absolute correspondence between the genotype of the xenografts and the MTAP staining by immunohistochemistry, with only mouse stromal cells staining positive for MTAP in *MTAP*-deleted xenografts (Supplementary Figs. 12–14). We thus proceeded to utilize this antibody for scoring homozygous *MTAP* deletions in human primary GBM FFPE sections. Immunohistochemistry was performed as described for the human xenografted tumors. Immunohistochemistry was performed by utilizing citrate antigen retrieval and blocking in 2% goat serum. Slides were stained against MTAP (Abcam, rabbit monoclonal, #ab126623, Lot: GR97816-5, Lot: YI070108C5, Lot: GR90092-13, overnight at 4 °C) in a 1:250 dilution and developed with 1× goat anti-rabbit IgG secondary, poly-horseradish peroxidase conjugate (Thermo Scientific, #B40962) for 30 min at room temperature and developed with NOVAred (Vector Laboratories) or EnzMet (Nanoprobes, #6001, 30 ml) followed by counterstaining with hematoxylin or hematoxylin and eosin. The same procedure was used to stain the slides against the IBA1 antibody (Abcam, #ab178846, lot: GR207976-27) in a 1:1000 dilution.

**Western blot**. Human snap-frozen GBM tumors were kept at −80 °C (same set of tumors used for metabolomic study). Without thawing, approximately 10 mg of tumor tissue was transferred to a new tube for immunoblotting. Lysates were made in 200 μl of radioimmunoprecipitation assay buffer supplemented with protease inhibitor (Roche, #11836153001) and phosphatase inhibitor (Roche, #04906837001) followed by sonication at 4 °C. Protein quantitation was performed via BCA assay (Pierce, #23225), and lysates were equilibrated for running on sodium dodecyl sulfate-polyacrylamide gel electrophoresis and transferred to a polyvinylidene fluoride membrane. The following antibodies were used for this study: SDMA motif (Cell Signaling Technology, #13222, Lot: 6) in a 1:1000

dilution, MTAP (Abcam, #ab126623, Lot: GR97816-5, Lot: YI070108C5, Lot: GR90092-13) in a 1:1000 dilution, GAPDH (Sigma-Aldrich, #G9545, lot: 127M4814V) in a 1:5000 dilution, IBA1 (Abcam, #ab178846, lot: GR207976-27) in a 1:1000 dilution, and vinculin (Cell Signaling Technology, #13901) in a 1:5000 dilution. Western blot bands were quantified using ImageJ 1.52q.

**Reporting summary**. Further information on research design is available in the Nature Research Reporting Summary linked to this article.

## Data availability

Metabolomic data and uncropped scans of all blots are provided in the Source data file. Raw NMR data in Fig. 4 and all metabolomic data are deposited in Figshare (https://doi.org/10.6084/m9.figshare.14608002.v5). Data in Supplementary Fig. 11a, b with file name 011210QssCSF2HGnormGBM.txt are kindly shared by John Asara using the BIDMC metabolomic platform, originally published in Locasale et al.[47]. Data in Supplementary Figs. 6g and 16d obtained from cBioPortal[42,43] (https://www.cbioportal.org) with source data are available at http://gdac.broadinstitute.org/runs/stddata__2016_01_28/data/GBM/20160128/ and Brennan et al.[41], respectively. Public domain metabolomic data used in Supplementary Figs. 2a–c, 4a, 6, 7, 9, and 11c–e are available at Ortmayr el al.[22], Su et al.[26], Dettmer et al.[27], Kryukov et al.[15], Prabhu et al.[33], Chinnaiyan et al.[44], Sanderson et al.[11], and Xiong et al.[48], respectively. Raw NMR data used in Supplementary Figs. 10e, 18, 19a, c, and 20 are available from the corresponding author upon reasonable request. Source data are provided with this paper.

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

## Acknowledgements

This work was financially supported by the following grants to F.L.M.: U.S. National Institutes of Health (NIH) grant 1R21CA226301, the American Cancer Society Research Scholar Award RSG-15-145-01-CDD, the National Comprehensive Cancer Network–Young Investigator Award YIA170032, and the Andrew Sabin Family Foundation Fellows Award. The University of Texas MD Anderson Cancer Center/Glio-blastoma Moon Shot, The CABI/GE In-Kind Research Grant (MI2), the Brockman Medical Research Foundation, and the SPORE in Brain Cancer (2P50CA127001) funds also supported the work. Y.B. was supported by the Schissler Foundation, Dr. John J. Kopchick, and Ms. Charlene Kopchick. S.K. was supported in part by MD Anderson Cancer Center CPRIT Research Training Program Grant RP170067. We thank Lisa Norberg and Kristin Alfaro-Munoz for GBM FFPE slide preparation and Edward Chang for slide scanning. The manuscript was edited by Sarah Bronson of the Research Medical Library at MD Anderson Cancer Center. We acknowledge Dr. Jason W. Locasale and Dr. Sydney M. Sanderson for Supplementary Fig. 9 data. We acknowledge Dr. Jason Cantor for sharing HPLM media. We also acknowledge Dr. Mark T. Bedford for helpful discussions. We thank Dr. Joshua D. Rabinowitz for providing his laboratory's help and resources for some MTA measurements in vitro. Illustrations are created with BioRender.com.

## Author contributions

Y.B. performed metabolomic sample preparation, NMR studies, and data analysis; J.J.A., S.K., E.B., and N.B.S. performed western blots; Y.B., J.J.A., K.A., K.-C.C., T.T., and A.H.P. performed immunohistochemistry; Y.-H.L. and Y.B. performed molecular biology and cell culture. Y.B. and D.K.G. generated figures; Y.B. and F.L.M. analyzed the data. J.T.H. analyzed IHC slides. J.d.G., J.T.H., A.d., and R.V. shared genomic data and primary tumor samples; J.M.A. and L.W. performed metabolomic analysis; R.K. provided the insightful edits; Y.B., V.C.Y., and F.L.M. conceived the study and; Y.B. and F.L.M. wrote the paper.

## Competing interests

The authors declare no competing interests.
