## [Peer Review File · Nature Communications]

REVIEWER COMMENTS

Reviewer #1 (Remarks to the Author); expert on metabolism:

In the paper entitled “Metabolomic profilings do not corroborate elevation of Methylthioadenosine in MTAP-deleted Glioblastoma” Barekattain and colleagues investigate the metabolic changes in cells deficient for the enzyme Methylthioadenosine Phosphorylase (MTAP), lost in a subset of glioblastoma tumours and cell lines, in combination with the tumour suppressor P16. They observe that, contrary to the expectations, MTAP-deficient cells exhibit only a modest intracellular accumulation of methylthioadenosine (MTA, the substrate of MTAP), and this metabolite is mostly released in the extracellular milieu. In addition, they propose that secreted MTA can be taken up by stromal immune cells, limiting the overall accumulation of this metabolite in situ.

These findings call for caution in inferring metabolic behaviour from tissue culture experiments, and for therapeutic intervention of MTAP-deficient tumours based on the expected accumulation of MTA. Although these results are relevant some concerns should be addressed before publication.

1) The authors propose that MTAP-deficient cells do not accumulate MTA in vivo is because its accumulation is a “predominantly extracellular phenomenon” and that stromal cells metabolize the secreted MTA. Yet, recent work (Sanderson et al (Science Advances 2019), has highlighted the lack of correlation between MTAP status and MTA accumulation and ascribed it to nutrient availability and heterogeneity in one-carbon metabolism. Regrettably, the authors discuss this important work only briefly at the end of the discussion. It is recommended that the authors introduce the work of Sanderson early on, given its relevance, and present the data in the context of these earlier findings. In addition, they could use the data shown in this paper to further validate their hypothesis and their analyses of external datasets. Finally, and more importantly, the authors should assess whether the accumulation and secretion of MTA in cells are confirmed using a more physiological medium (Plasmax or HPLM), where the concentration of metabolites that affect methionine pathway reflects those found in the blood. Indeed, it is possible that the lack of accumulation of MTA in MTAP-deficient tumours is due to the depletion of specific nutrients, including methionine and cysteine, in vivo, rather than by secretion and scavenging. Using a physiological medium would allow a more faithful comparison between their observation in vitro and vivo, providing strength to the central hypothesis.

2) In Fig 2 B-C the authors compare the levels of MTA in cells and media and conclude that the “modest intracellular increases of MTA were vastly overshadowed by extracellular levels of MTA, again evidencing secretion of MTA”. Yet, it can be misleading to compare ion intensities of MTA between cells and media, given that the concentration of the extracts (and how they are prepared) is likely different. To corroborate this conclusion, the authors should calculate the absolute concentration of MTA in cells and media, normalising for volume of extract, volume of media, and cell volume.

3) In Fig 3F the authors show the level of SDMA in MTAP positive and negative tumours. However, based on MTAP level, only one MTAP-deficient tumour shows genuine loss of MTAP protein levels, casting doubts about their classification and the validity of the conclusions made from these experiments. Although this could be due to stromal contaminations of MTAP positive cells, can the authors clarify this issue?

4) Fig 1 is a schematic representation of methionine salvage and the proposed mechanism of regulation of PRMT5 by MTA. This figure can be moved to supplementary material.

Reviewer #2 (Remarks to the Author); expert on GBM:

Barekattain and colleagues perform analysis of previously published and new metabolic profiling studies to suggest that excess MTA in MTAP-deleted gliomas is secreted into the extracellular environment and processed by MTAP-intact cells within the tumor microenvironment, leading them to propose that MTAP-deletion/excess MTA may not be a good therapeutic target. Homozygous deletion of 9p21 is frequent in GBM, leading in deletion of well-known tumor suppressor CDKN2A and collateral deletion of MTAP, which has provided prior motivation for targeting this vulnerability.

MTAP has a role in adenine and methionine salvage pathways and deletion of MTAP leads to elevated MTA in many in vitro models; there are now strategies under investigation that exploit the elevation in MTA (PRMT5i and MAT2Ai). The authors note that despite much in vitro data, there is currently no evidence to support that MTA levels are elevated in MTAP deleted tumors, such as primary GBM. Further, upon analysis of published metabolomics data, they observe that MTA elevation is primarily extracellular and there are no tumors with extreme MTA elevation.

The authors analyzed a panel of 18 primary GBM for MTA and SAM levels and observe no significant difference in levels between MTAP intact and low MTAP expressing tumors, though MTA levels are generally higher in the lines with lower MTAP expression. They demonstrate that MTAP staining is intact in MTAP-WT and stromal tissue in human samples while absent in MTAP-deleted tumors. This is also true in xenografts. Having demonstrated that in vivo MTAP is present in the tissues surrounding the cell, the authors show by co-culture of MTAP-deleted and MTAP-intact cell lines that elevated MTA in the media of MTAP-deleted lines can be cleared from the media when MTAP-WT cells are present. As an indirect measure of PRMT5 activity (which is inhibited by elevated MTA), PRMT5 activity increases, leading to increased levels of SDMA, with co-culture leading to MTA clearance from the media.

Overall, this is an interesting study but there are a couple critical issues that need to be addressed to make the data more convincing. Specifically, the authors need to strengthen data surrounding the lack of correlation between MTAP and MTA in vivo. MTA elevations have previously been reported in MTAP-deleted tumors and I am not convinced by how they defined MTAP-deleted in their GBM sample population. Theoretically, MTAP-deleted tumors should exhibit no MTAP expression by western blot, and there are certainly patient-derived GBM glioma models in use that do not have detectable protein levels of MTAP. Here, however, the authors are using MTAP-deletion and low MTAP interchangeably but they have not provided sufficient evidence that this is appropriate. This makes me wonder if a significant correlation between MTAP loss and MTA elevation might be found if their definition/threshold for “low MTAP expression” was more stringent.

- MTAP-deletion and “low MTAP expression” are not necessarily the same and I worry that the authors assumption of this leads to inaccurate conclusions. How exactly was MTAP expression measured in Supplementary Figure 2 and what is the label of the y-axis. Is this protein level (western blot/IHC), mRNA, something else? mRNA level is not always equivalent to protein abundance. A tumor with homozygous deletion of 9p21 would be CN = 0 and therefore no expression at all.
- Along these lines, MTAP expression is observed in the tumors that the authors are calling “MTAP deleted”. Low (but present) MTAP protein level may be sufficient to regulate MTA levels. In Figure 3, it would be helpful to see how MTA levels correlate with amount of MTAP protein present by western blot.
- Data in Supplementary Figure 4 do not support your other data – these show that MTA levels (normalized to mean or normalized to SAM) are significantly higher in MTAP-deleted tumors. Why the discrepancy? And, are these tumor truly MTAP-deleted or MTAP-low?
- Supplementary Figure 6D and its conclusions are based on assumptions and not known genotypes of the patients being sampled. This is a weak argument.
- Figure 1A is not data but summary of the pathway – perhaps the TCGA data should be omitted since this is not primary data.
- It is difficult to read the y-axis units on Figure 2C – is this 10Z?
- Please be cautious in the discussion of MAT2A as a target. Though MAT2A's important cellular role

is noted, many targeted therapies are aimed at essential genes (ex EGFR null mice have early lethality) – it is the therapeutic window that is attempted to be exploited with MAT2A inhibition in the setting of MTAP deletion.

- The GBM tumors label is out of place on Suppl. Fig 3.

Reviewer #3 (Remarks to the Author); expert on MTAP-deleted targeting:

A major conclusion from this paper is that MTA is excreted by GMB cells and is metabolized by stromal cells.

Evidence for products of metabolism of MTAP would further strengthen their conclusions,

Reviewer Comments:

Reviewer #1 (Remarks to the Author); expert on metabolism:

In the paper entitled “Metabolomic profilings do not corroborate elevation of Methylthioadenosine in MTAP-deleted Glioblastoma” Barekattain and colleagues investigate the metabolic changes in cells deficient for the enzyme Methylthioadenosine Phosphorylase (MTAP), lost in a subset of glioblastoma tumours and cell lines, in combination with the tumour suppressor P16. They observe that, contrary to the expectations, MTAP-deficient cells exhibit only a modest intracellular accumulation of methylthioadenosine (MTA, the substrate of MTAP), and this metabolite is mostly released in the extracellular milieu. In addition, they propose that secreted MTA can be taken up by stromal immune cells, limiting the overall accumulation of this metabolite in situ. These findings call for caution in inferring metabolic behaviour from tissue culture experiments, and for therapeutic intervention of MTAP-deficient tumours based on the expected accumulation of MTA. Although these results are relevant some concerns should be addressed before publication.

1) The authors propose that MTAP-deficient cells do not accumulate MTA in vivo is because its accumulation is a “predominantly extracellular phenomenon” and that stromal cells metabolize the secreted MTA. Yet, recent work (Sanderson et al (Science Advances 2019), has highlighted the lack of correlation between MTAP status and MTA accumulation and ascribed it to nutrient availability and heterogeneity in one-carbon metabolism. Regrettably, the authors discuss this important work only briefly at the end of the discussion. It is recommended that the authors introduce the work of Sanderson early on, given its relevance, and present the data in the context of these earlier findings. In addition, they could use the data shown in this paper to further validate their hypothesis and their analyses of external datasets. Finally, and more importantly, the authors should assess whether the accumulation and secretion of MTA in cells are confirmed using a more physiological medium (Plasmax or HPLM), where the concentration of metabolites that affect methionine pathway reflects those found in the blood. Indeed, it is possible that the lack of accumulation of MTA in MTAP-deficient tumours is due to the depletion of specific nutrients, including methionine and cysteine, in vivo, rather than by secretion and scavenging. Using a physiological medium would allow a more faithful comparison between their observation in vitro and vivo, providing strength to the central hypothesis.

⇒ The reviewer makes two points which we have addressed experimentally: A) How the results of Sanderson *et al.*¹ apply to our present work and B) whether the extracellular accumulation of MTA in MTAP-deleted cells is still evident in physiological media.

⇒ In response to point A: The reviewer's comment is spot on—we should indeed have discussed the seminal paper by Sanderson *et al.*¹ more deeply. Sanderson *et al.*¹ performed a deep metabolomic investigation into the factors that affect MTA levels in *MTAP*-deleted vs. intact, including nutrient concentrations, amino acids, etc. We should begin by stating that we do not explicitly rule out the hypothesis that nutrient differences (i.e., extremely low cysteine levels as shown by Sanderson *et al.*¹) may contribute to the discrepant observation of lack of MTA accumulation in primary human tumors with *MTAP* homozygous deletions. Sanderson *et al.*¹ show that *in vitro*, the availability of methionine significantly modulates the levels of MTA (lower methionine => low MTA), but yet, the relative levels of MTA are still higher in *MTAP*-deleted vs. *MTAP*-intact cancer cells when grown in restricted methionine medium (**Supplementary Figure S9** from Sanderson *et al.*¹). We thus asked whether methionine levels would correlate with MTA levels in primary GBM tumors and whether *MTAP* deletion status would now emerge as a significant separator if methionine levels are taken as a covariate. Analysis of two independent metabolomic datasets from primary GBM tumors with known *MTAP*-deletion status indicates that MTA levels correlate with Methionine levels in primary tumors. However, no difference between *MTAP*-deleted and *MTAP*-intact genotype emerges (**The Below Figure**). At the same time, MTA levels positively correlate not just with methionine, but with most other amino acids (including serine) and the sum of all amino acids, with higher statistical significance than methionine. This suggests that whatever association between methionine and MTA is unlikely to be specific and causative and more likely reflects the cellularity of individual tumors.

A**MDA series (Metabolon Inc.)****MTAP-deleted and intact tumors**

Pearson correlation coefficient

$-\log_{10}(\text{P value})$

All amino acids

Cys

Met

B**HF series (BIDMC)****MTAP-deleted and intact tumors**

Pearson correlation coefficient

$-\log_{10}(\text{P value})$

Met

All amino acids

Cys

MTA levels correlate with amino acids in primary GBM tumors in two independent metabolomic datasets (**A**) MDA series with N=9 *MTAP*-deleted and N=14 intact tumors, and (**B**) HF series with N=9 *MTAP*-deleted and N=8 *MTAP*-intact. MTA levels for the individual tumors in each dataset are shown in **Supplementary Figure S5** (MDA series) and **Figure 2** (HF series). A positive correlation was observed between levels of MTA and methionine. However, this correlation is also observed between MTA and many other individual amino acids, as well as the summation of all amino acids. This conclusion stands whether correlations of MTA levels with amino acids are interrogated in *MTAP*-deleted and *MTAP*-intact tumors separately (upper and middle panels) or pooled (lower panels).

⇒ As the reviewer suggested, we took a deep dive analysis of Sanderson *et al.*¹ paper; we re-plot this data for glioma cell lines in **Supplementary Figure S9**. Their data suggest methionine or cysteine restricted media do indeed profoundly modulate MTA levels. We note that MTA levels decreased in *MTAP*-deleted and WT cells cultured in extremely low methionine conditions. However, the

difference in MTA levels between the *MTAP*-deleted cells and WT remains significant—more MTA in *MTAP*-deleted cells than WT. Their data suggest that MTA accumulated (although less) inside *MTAP*-deleted cells even when cells were cultured in the restricted methionine condition. While methionine restriction can lower MTA levels in cells, it could not explain why we do not observe the significant difference in MTA levels between *MTAP*-deleted and intact human GBM tumors. On the other hand, Sanderson *et al.*¹ data show that culturing cells in the restricted cysteine media results in decreased MTA levels only in *MTAP*-deleted glioma cells, but not WT cells. This may indicate that cysteine deprivation inside tumors could cause no significant MTA accumulation in *MTAP*-deleted human GBM tumors. However, we note two points (i) the cysteine and cystine's average concentrations in human plasma (34 μ M and 48 μ M) are much higher than their concentration in the cysteine restricted condition (0 μ M and 6 μ M); and (ii) Sanderson *et al.*¹ used a shorter culture time (6 hours) for the restricted cysteine experiments due to the toxicity caused by the cysteine restriction. Since the accumulation of MTA is time-dependent (more MTA accumulates with increasing culture time, **Figure 1F**), it could be possible that the difference in MTA levels between *MTAP*-deleted and WT cells cultured in restricted cysteine conditions became more significant if cells were grown for a longer time (i.e., 24 hours). As mentioned in our paper, we do not explicitly rule out the possibility that nutrient differences cause the discrepant observation of lack of MTA accumulation in primary human tumors with *MTAP* deletions.

Supplementary Figure S9

Data re-plotted from Sanderson *et al.*¹³

Supplementary Figure S9: Disparate effects of methionine and cysteine deficiency on MTA levels in MTAP deleted versus intact cancer cells. Data re-plotted from Sanderson *et al.*, Science Advances, 2019¹³. The comparison of MTA levels in the MTAP-deleted and WT glioma cells when cells cultured in the standard formulation of RPMI (100 μ M methionine and 200 μ M cysteine) versus restricted methionine (3 μ M) or restricted cysteine (6 μ M) conditions. MTA levels decreased in MTAP-deleted and WT cells when cells were cultured in extremely low methionine conditions. However, the difference in MTA levels between the MTAP-deleted cells and WT remains significant – more MTA in MTAP-deleted cells than WT. This data suggest that MTA accumulated inside MTAP-deleted cells even when cells were cultured in the restricted methionine condition. While methionine deficiency can lower MTA levels in cells but could not explain why we do not observe the significant difference between MTAP-deleted and WT human GBM tumors. On the other hand, culturing cells in the restricted cysteine media decreased MTA levels only in the MTAP-deleted glioma cells, but not MTAP-WT cells. This may indicate that cysteine deprivation inside tumors could potentially cause no MTA accumulation in MTAP-deleted human GBM tumors. However, we note that cysteine and cystine's average concentrations in human plasma (34 μ M and 48 μ M) are much higher than their concentration in the cysteine restricted condition (0 μ M and 6 μ M). **P=0.02, significant values are obtained using multiple t-test analysis with Benferroni correction.

⇒ As discussed already, the relative difference in MTA levels between *MTAP*-deleted and intact cancer cells is still significant when comparing cells grown in restricted versus complete methionine conditions even though absolute MTA levels are lower (**Supplementary Figure S9**, Sanderson *et al.*¹ data). However, such difference is not significant when comparing MTA levels between *MTAP*-deleted cells cultured in the restricted methionine (3 μ M Met) vs. *MTAP*-intact cells grew in complete methionine (100 μ M Met). Thus, we sought to see if methionine levels differ significantly between *MTAP*-deleted GBM tumors and intact ones (**The Below Figure**). Interrogating two independent metabolomic data sets (different sets of tumors and independent

mass-spect cores), we did not observe any significant difference (increase or decrease) in levels of methionine as well as cysteine and serine between *MTAP*-deleted and intact human GBM tumors. This indicates that the variation between methionine and cysteine levels between *MTAP*-deleted and intact GBM tumors is minimal. Moreover, to see if the levels of methionine and cysteine in GBM tumors are extremely lower than brain, we compared the levels of these metabolites in a mouse brain (as an illustrative of a human brain since we do not have any normal human brains for such comparisons) with primary GBM tumors from the same dataset (**The Below Figure, Panel A**). We did not observe methionine and cysteine levels to be significantly lower in GBM than in the normal brain.

There is no significant difference in levels of methionine and cysteine between *MTAP*-deleted GBM tumors and intact ones. The methionine, cysteine, and serine levels in two independent datasets (**A**) MDA series using Metabolon Inc. platform and (**B**) HF series using BIDCM platform. *MTAP*-deletion status of tumors is determined as discussed in the manuscript (**Figure 2** and **Supplementary Figure S5**) and this document. No significant difference was observed in methionine and cysteine levels in *MTAP*-deleted and intact tumors. Moreover, these metabolite levels are not significantly higher in normal mouse brain (representing normal human brain) and GBM tumors.

⇒ In response to point B, we repeated the cell culture experiments using a more physiological medium (Plasmax), as the reviewer suggested. Like our *in vitro* experiments performed in the

traditional cell culture medium (DMEM), we observed modest intracellular accumulation of MTA in *MTAP*-deleted cells, and significant extracellular accumulation of MTA in *MTAP*-deleted conditioned media compared to *MTAP*-WT cells (**Supplementary Figure S10B-E**). Extracellular accumulation of MTA was broadly similar in *MTAP*-deleted cells cultured in DMEM (supraphysiological cysteine/methionine) versus those in Plasmax (physiological cysteine/methionine). We also compared SDMA (an index of PRMT5 activity, decreased by MTA) levels of *MTAP*-deleted and WT cells cultured in DMEM vs. Plasmax and HPLM. We found no significant difference in the SDMA levels of cells cultured in traditional supraphysiological media vs. physiological ones (**Supplementary Figure S10F**). We also performed the co-culture experiments in Plasmax. **Supplementary Figure S18D** confirms MTA's secretion by homozygous *MTAP*-deleted cells and its scavenging by *MTAP*-WT macrophages in Plasmax.

Supplementary Figure S10

Supplementary Figure S10: Comparison of intracellular and extracellular MTA levels in cancer cells cultured in DMEM versus Plasmax. (A) Comparison between the concentrations of methionine, cysteine, and cystine in historic medium (DMEM) and physiological media (Plasmax and HPLM) and human plasma. The formulations of physiological media such as Plasmax better recapitulate the human plasma and tumors' nutrition environment. Historic media contain a much higher concentration of some metabolites, i.e., methionine and cystine, compare to human plasma, but they lack many other metabolites, i.e., cysteine. *MTAP*-deleted and WT cells were cultured in DMEM, and Plasmax supplemented with 2.5% FBS. After three days, cells and media were harvested for mass-spectroscopy, NMR, and western blotting. DMEM, Plasmax, and HPLM are indicated as D, P, and H, respectively. (B and C) Intracellular and extracellular levels of MTA of cells cultured in DMEM vs. Plasmax using mass spectrometry. The average MTA levels in the cell pellet or conditioned media of the *MTAP*-deleted cells do not differ significantly between DMEM and Plasmax. (D) Absolute quantification of MTA in cell pellet and conditioned media of cells cultured in Plasmax. The amount of MTA recovered from conditioned media of *MTAP*-deleted cells cultured in Plasmax is significantly greater than that recovered from the cell pellets, recapitulate the cell culture data in DMEM. Thus, *MTAP*-deleted cells cultured in Plasmax secrete MTA to the conditioned media resulting in the predominant extracellular presence of MTA. (E) Further validation of accumulation extracellular MTA in DMEM vs. Plasmax by NMR. (F) SDMA levels of *MTAP*-deleted and WT glioma cells when cultured in DMEM vs. Plasmax and HPLM. No significant difference is observed in the SDMA levels of cells cultured in traditional medium and physiological ones.

Supplementary Figure S18D

Supplementary Figure S18D: Co-culture of macrophages with *MTAP*-deleted cancer cells abrogates MTA accumulation in conditioned media. Secretion by *MTAP*-deleted and scavenging of MTA by *MTAP*-WT cells are further confirmed in physiological medium (Plasmax). *MTAP*-deleted cells were cultured with and without macrophage cells (RAW-264.7) for 3 days in DMEM or Plasmax supplemented with 2.5% FBS. This figure illustrates that secreted MTA by *MTAP*-deleted cells can further metabolize by *MTAP*-intact macrophages regardless of cell culture media (DMEM vs. Plasmax), abrogating MTA accumulation in the extracellular environment.

2) In Fig 2 B-C the authors compare the levels of MTA in cells and media and conclude that the “modest intracellular increases of MTA were vastly overshadowed by extracellular levels of MTA, again evidencing secretion of MTA”. Yet, it can be misleading to compare ion intensities of MTA between cells and media, given that the concentration of the extracts (and how they are prepared) is likely different. To corroborate this conclusion, the authors should calculate the absolute concentration of MTA in cells and media, normalising for volume of extract, volume of media, and cell volume.

⇒ We agree with the reviewer’s points and have performed additional experiments to measure the absolute amounts of MTA in cell pellet versus media. To do so, in collaboration with J Rabinowitz’s lab, we performed new experiments and measured the absolute amounts of MTA

(nanomoles) found in the cell pellets and paired conditioned media (**Figure 1H**). For these experiments, the entire cell pellet was extracted; an aliquot of media was extracted, the concentration was determined using deuterium-labeled MTA as an internal standard. The concentration of MTA in conditioned media was corrected for the percentage of media to extraction buffer. The absolute amounts of MTA in cell pellet and media were determined by multiplying the concentration by the total volume of cell and media extract.

The key points from these comparisons are:

- I. Conditioned media has dramatically higher absolute amounts of MTA in *MTAP*-deleted (24.637 ± 5.357 nanomoles, mean \pm SD) vs. *MTAP*-intact cells (0.006 ± 0.003 nanomoles, mean \pm SD). This can also be seen as ion count levels in **Figure 1E**.
- II. Cell pellets show the same trend but of a less magnitude — only a 6-fold increase in intracellular MTA in *MTAP*-deleted vs. *MTAP*-intact cells. The absolute concentration of MTA in cell pellet of *MTAP*-deleted and intact cells are 0.113 ± 0.036 nanomoles and 0.017 ± 0.006 nanomoles (mean \pm SD), respectively.
- III. The amount of MTA in conditioned media (extracellular) is dramatically higher than in the cell pellet (intracellular) in *MTAP*-deleted cells (MTA in media $\sim 24.637 \pm 5.357$ nanomoles vs. cell pellet 0.113 ± 0.036 nanomoles, mean \pm SD). On the other hand, there is slightly more MTA present in the cell pellet of *MTAP*-WT cell pellet versus conditioned media (MTA in media 0.006 ± 0.003 nanomoles vs. cell pellet 0.017 ± 0.006 nanomoles, mean \pm SD). Together, these data indicate *MTAP*-deleted cells expel excess MTA into the extracellular milieu.

Figure 1H

Figure 1H: Absolute quantification of MTA in cell pellet and conditioned media. The amount of MTA recovered from conditioned media of *MTAP*-deleted cells is 200-fold greater than recovered from the cell pellets. In contrast, in *MTAP*-WT cells, the amount of MTA in conditioned media is comparable to that in cell pellet. The modest intracellular accumulation of MTA (6-fold) in the *MTAP*-deleted cells compared to the WT cells is overshadowed by the extraordinary increase of extracellular MTA. Together, these data emphasize that *MTAP*-deletion predominantly promotes extracellular rather than intracellular MTA accumulation.

⇒ A similar conclusion was reached by Shlomi *et al.*,²; the authors found significantly higher MTA secretion in an *MTAP*-deleted cell line than WT.

Shlomi et al., Analytical Chemistry 2014

Table 1. Inferred Fluxes in nmol/ μ L-cells/h in Both HT1080M- and HT1080M+ Based on Isotope Labeling Coupled with Computational Modeling^a

reaction	HT1080M+	HT1080M-
$v_1 (v_{met_in} - v_{met_out})$	0.8 \pm 0.1	0.9 \pm 0.1
v_2	0.17 \pm 0.02	0.11 \pm 0.01
v_3	0.17 \pm 0.02	0
v_4	0.12 \pm 0.02	0.16 \pm 0.02
v_5	0.03 \pm 0.02	0.02 \pm 0.01
v_6	0	0
v_7	0.18 \pm 0.01	0.33 \pm 0.03
v_8	0.07 \pm 0.01	0.13 \pm 0.01
v_9	0.11 \pm 0.01	0.2 \pm 0.02
v_{MTA}	<0.005	0.11 \pm 0.01
v_{hyc}	0.09 \pm 0.01	0.14 \pm 0.02
v_{putr}	0.01 \pm 0	0.21 \pm 0.02
v_{pr_net}	0.71 \pm 0.1	0.65 \pm 0.02

Table 1 from Shlomi *et al.* paper indication significantly higher MTA secretion in the *MTAP*-deleted cell line than WT, supporting our data regarding MTA's secretion by *MTAP*-deleted cells.

3) In Fig 3F the authors show the level of SDMA in *MTAP* positive and negative tumours. However, based on *MTAP* level, only one *MTAP*-deficient tumour shows genuine loss of *MTAP* protein levels, casting doubts about their classification and the validity of the conclusions made from these experiments. Although this could be due to stromal contaminations of *MTAP* positive cells, can the authors clarify this issue?

⇒ Human GBM tumors are made up of a mix of malignant glioma (cancer) cells with genetic alterations like homozygous *MTAP*-deletion and non-transformed non-malignant stromal cells that do not carry any genetic abnormalities. Thus, even in homozygous *MTAP*-deleted tumors, only malignant glioma cells lack *MTAP*, while stromal cells express normal levels. Approximately 25 to 75% of cell mass in a GBM tumor are *MTAP*-expressing stroma (e.g., endothelial, immune, glial, and stem cells)³⁻⁵. Because such non-malignant stromal cells make up a substantial

Figure from Behnan *et al.*, Brain, 2019. GBM tumors consist of malignant, transformed, and mutated glioma cells and non-malignant non-mutated stromal cells. Glioma cells in homozygous *MTAP*-deleted GBM tumors do not express *MTAP*. However, stroma cells are *MTAP*-WT and express *MTAP*. As a result, when a bulk resected GBM tumor is probed by Western blot and mRNA expression, *MTAP* protein is still expressed even if the glioma cells are homozygous *MTAP*-deleted and non-expressing.

proportion of GBM mass and express WT levels of MTAP, there is no expectation that a western blot performed on a lysate of the bulk resected tumor (containing an admixture of malignant *MTAP*-null cancer cells and *MTAP*-expressing stromal cells) would show a complete absence of *MTAP*.

⇒ On average, the *MTAP*-deleted tumors express a lower amount of *MTAP* protein levels by the western blot than *MTAP*-intact (**below graph**). However, some expression of *MTAP* deriving from *MTAP*-WT stromal cells will always remain. This is why immunohistochemistry is necessary and allows clear discrimination of the residual expression of *MTAP* in

Figure 2H

Figure 2H: *MTAP* protein levels corrected for loading with GAPDH control from the western blot of HF series tumors shown in **Figure 2G**. Due to the presence of *MTAP*-WT cells in the lysate of whole tumors, *MTAP* protein levels are not zero even for homozygous *MTAP*-deleted tumors. However, on average homozygous *MTAP*-deleted tumors have lower *MTAP* protein levels compared to intact tumors.

non-transformed stromal cells versus its absence in glioma cells. To illustrate that stroma exists in bulk resected tumors and lysates derived from it, we immunoblotted myeloid marker (IBA1, as microglia/macrophages constitute the largest tumor stroma component in GBM, **Figure 2G**, and **Supplementary Figure S5**). All tumors show some levels of IBA1 expression, which in *MTAP*-

Figure 2G

Figure 2G: *MTAP* protein levels in bulk tumors are lower in *MTAP*-deleted compared to *MTAP*-intact ones but are not zero due to the presence of *MTAP*-expressing non-malignant stromal cells (e.g., microglia, astrocytes, and endothelial cells). The presence of microglia in a tumor is confirmed by the microglial markers IBA1 in tumor lysates but not seen in cells in culture.

The non-zero levels of *MTAP* protein for the homozygous *MTAP*-deleted HF3174 tumor (marked with red arrow) is more evident in higher exposed band of *MTAP*. In contrast, no *MTAP* expression is detected in the lysate of *MTAP*-deleted cells (U87).

deleted tumors, nicely correlates with residual MTAP. Interestingly, the reviewer mentioned that the *MTAP*-deleted HF3174 tumor (a red arrow in **Figure 2G**) as the “only one *MTAP*-deficient tumour shows genuine loss of *MTAP* protein levels.” Note the lower levels of IBA1 protein (low myeloid stroma infiltration) for this tumor (HF3174), resulting in low but non-zero expression of *MTAP* compared to other homozygous *MTAP*-deleted tumors.

⇒ IHC further validates the presence of myeloid (macrophage/microglia) stromal cells in GBM tumors with a strong correlation between residual *MTAP* IHC staining and IBA1-positive myeloid (macrophage/microglia) content in homozygous *MTAP*-deleted human GBM tumors (**Supplementary Figure S16**). In homozygous *MTAP*-deleted tumors, the residual *MTAP* staining is restricted to stromal cells (no *MTAP* staining in malignant glioma cells), predominantly of myeloid origin (IBA1-positive). While myeloid cells (microglia/macrophages) constitute the most

Supplementary Figure S16

Supplementary Figure S16: Myeloid cells constitute the majority of *MTAP*-expressing stromal cells in *MTAP*-deleted GBM tumors. FFPE serial sections were stained against *MTAP* (first column) and IBA1 (myeloid marker, stains macrophages/microglia, second column) and images acquired from consecutive sections of the same tumor. (A) Case 503710, *MTAP*-deleted; (B) Case 997730, *MTAP*-deleted; (C) Case 506330, *MTAP*-deleted. Human GBM tumors may have up to 75% non-malignant stromal cell content, including *MTAP*-intact microglia. The presence of the *MTAP*-expression myeloid cells results in *MTAP*-positive stains even in GBM tumors that are homozygous *MTAP*-deleted, since only the malignant glioma cells carry the deletion.

significant component of tumor stroma in GBM, there are other *MTAP*-WT non-malignant cells such as glial, endothelial, and stem cells, which contribute to the *MTAP* protein levels detected in Western blot/IHC. More *MTAP* IHC staining for human xenografts in mouse and primary GBM tumors is shown in **Figure 3** and **Supplementary Figures S12-S17**, in which *MTAP*-expressing stroma cells in homozygous *MTAP*-deleted tumors stained strongly, while malignant glioma cells did not.

⇒ We also interrogate an independent data set (TCGA⁶) for the correlation between mRNA levels of MTAP and AIF1 (the official gene symbol of IBA1, microglia/macrophages marker) among GBM tumors with known *MTAP*-status. **Supplementary Figure S16D** illustrates that homozygous *MTAP*-deleted GBM tumors (Blue dots), on average, have lower but non-zero levels of MTAP mRNA levels compared to *MTAP*-intact tumors (Gray squares), consistent with our finding from the Western blot on bulk tumors (**Figure 2G**). This figure further validates that IBA1-positive myeloid cells (which constitute most stroma in GBM) drive the non-zero MTAP expression in bulk *MTAP*-deleted GBM tumors and indicates that there is a significant positive correlation between mRNA levels of MTAP and AIF1 in *MTAP*-deleted tumors vs. intact ones. However, no positive correlation was observed between MTAP and AIF in *MTAP*-intact tumors since both glioma, and stromal cells express MTAP in *MTAP*-intact tumors.

Supplementary Figure S16D

Supplementary Figure S16D: The correlation between MTAP mRNA levels and AIF1 (microglia/macrophages marker) mRNA levels for homozygous *MTAP*-deleted (blue) and *MTAP*-intact (gray) human GBM tumors. Data obtained from TCGA cell 2016⁶. GBM tumors are an admixture of cancer cells and stromal cells. Due to the presence of MTAP-expressing stromal cells such as microglia, homozygous *MTAP*-deleted GBM tumors express non-zero levels of MTAP. On average, homozygous *MTAP*-deleted tumors have lower levels of MTAP mRNA compared to *MTAP*-intact, and the stronger correlation between IBA1 and MTAP for homozygous *MTAP*-deleted GBM tumors.

⇒ In sum, the non-zero MTAP expression seen in Western blots of homozygous *MTAP*-deleted human tumors is due to the presence of non-malignant MTAP-expressing stromal cells in bulk tumors. The malignant glioma cells in homozygous *MTAP*-deleted tumors do not express MTAP.

⇒ Moreover, the reviewer had concerns regarding how we identified and classified homozygous *MTAP*-deleted tumors in our study. Homozygous *MTAP*-deleted and intact GBM tumors in **Figure 2** (HF series) were determined by array cGH (array comparative genomic hybridization) in the studies described by Kim *et al.*⁷ and Wang *et al.*⁸ In sum, DNA was isolated from whole tumors and hybridized with SNP 6.0 affix arrays. Copy numbers were corrected for tumor cellularity and whole-genome duplications. The *MTAP* deletion data (**Figure 2B**) was obtained from the publicly deposited data from Kim *et al.*⁷ We further confirmed *MTAP*-status of tumors by

Figure2B: Genomic copy-number data (dark blue: homozygous deletion, dark red: amplification) around the 9p21 locus. Each strip in the y-axis represents a single tumor at the specific chromosomal location (x-axis). Tumors with heterozygous *MTAP*-deletion and *MTAP*-WT tumors are classified as *MTAP*-intact in our manuscript.

immunohistochemistry against *MTAP* using a validated antibody (as described in the manuscript). *MTAP* status assignment of the individual tumors is not controversial and agrees with the *CDKN2A* status assignment by Kim *et al.*⁷

Figure 3A Kim et al., Genome Res 2015
(The assignment of *CDKN2A* deletion status)

Assignment of *CDKN2A* deletion status of individual tumors in the HF series by Kim *et al.* Genome Res 2015. The co-deletion of *MTAP* may be observed in 80%-90% of all tumors harboring the homozygous deletion of *CDKN2A*⁶. Except for HF2898 and HF3041, all homozygous *CDKN2A*-deleted tumors harbor homozygous *MTAP*-deletion as well. Tumors with heterozygous *MTAP*-deletion and *MTAP*-WT tumors are classified as *MTAP*-intact in our manuscript.

⇒ In sum, the *MTAP* status of tumors in **Figure 2** (HF series) was determined from previous cGH studies^{7,8} and the absence of *MTAP* expression in glioma cells verified further by IHC against *MTAP* using an antibody validated on *MTAP*-deleted xenografts (**Supplementary Figure S12-S14**). Malignant glioma (cancer) cells in homozygous *MTAP*-deleted tumors do not express any levels of *MTAP*. Tumors in which malignant glioma cells express low but non-zero levels of *MTAP*, such as heterozygous *MTAP*-deleted tumors are classified as *MTAP*-intact since only homozygous *MTAP*-deleted cells accumulate MTA in culture (**The Below Figure**, data re-plotted from Kryukov *et al.*⁹).

⇒ Moreover, in the revised version of this manuscript, we have performed and included additional experimental data to confirm further our findings of no significant elevation of MTA in homozygous *MTAP*-deleted GBM tumors (**Supplementary Figure S5**). The new metabolomic data were generated using an additional set of human GBM tumors (MDA series) and profiling with a different metabolic platform (Metabolon, Inc.). The *MTAP* status of tumors was confirmed by immunohistochemistry using a validated *MTAP* antibody. Consistent with findings from **Figure 2**, we did not observe a statistically significant increase in MTA levels between homozygous *MTAP*-deleted and intact human GBM tumors, verifying our initial observation.

Supplementary Figure S5

Supplementary Figure S5: No significant elevation of MTA in *MTAP*-deleted primary resected GBM tumors in an independent metabolomic data set. (A) Flash-frozen primary resected GBM tumors (heterogeneous mix of transformed glioma cells and non-malignant stromal cells) were prepared for IHC, mass spectrometry, and Western blotting. (B) *MTAP* deletion status of tumors was confirmed by immunohistochemistry using a validated *MTAP* antibody. Representative cases of *MTAP* positive and *MTAP* negative GBM tumors are shown. Non-malignant *MTAP*-intact cells are stained positive in the *MTAP*-deleted case. (C) MTA levels (absolute ion counts) in *MTAP*-deleted (blue) versus *MTAP*-intact (gray) GBM tumors, using the Metabolon, Inc. platform. Each bar represents MTA levels for each tumor. There is a 1.14-fold increase in the median MTA levels in *MTAP*-deleted tumors compared to intact tumors ($p = 0.9$, unpaired 2-tailed t-test with unequal variance). (D, E) Same data but expressed as a ratio of each tumor's total ion count for sample loading normalization (D) and as a ratio to SAM levels to account for upregulation in the methionine salvage pathway for each tumor (E). The median MTA/sum and MTA/SAM fold change in *MTAP*-deleted tumors compared to intact tumors are 1.23 ($p=0.9$) and 0.8 ($p=0.3$) respectively. (F) The levels of vinculin, *MTAP*, and IBA1 (microglia marker) in some of the tumors. Non-malignant *MTAP*-expressing cells (e.g., IBA1-expressing cells) drive the no-zero *MTAP* expression in the Western blot of homozygous *MTAP*-deleted GBM tumors.

4) **Fig 1 is a schematic representation of methionine salvage and the proposed mechanism of regulation of PRMT5 by MTA. This figure can be moved to supplementary material.**

⇒ In the revised manuscript, the methionine salvage representation is placed in **Figure 1A**. Moreover, the proposed mechanism of regulation of PRMT5 by MTA moved to the supplementary information (**Supplementary Figure S1**).

Reviewer #2 (Remarks to the Author); expert on GBM:

Barekattain and colleagues perform analysis of previously published and new metabolic profiling studies to suggest that excess MTA in MTAP-deleted gliomas is secreted into the extracellular environment and processed by MTAP-intact cells within the tumor microenvironment, leading them to propose that MTAP-deletion/excess MTA may not be a good therapeutic target. Homozygous deletion of 9p21 is frequent in GBM, leading in deletion of well-known tumor suppressor CDKN2A and collateral deletion of MTAP, which has provided prior motivation for targeting this vulnerability.

MTAP has a role in adenine and methionine salvage pathways and deletion of MTAP leads to elevated MTA in many in vitro models; there are now strategies under investigation that exploit the elevation in MTA (PRMT5i and MAT2Ai). The authors note that despite much in vitro data, there is currently no evidence to support that MTA levels are elevated in MTAP deleted tumors, such as primary GBM. Further, upon analysis of published metabolomics data, they observe that MTA elevation is primarily extracellular and there are no tumors with extreme MTA elevation. The authors analyzed a panel of 18 primary GBM for MTA and SAM levels and observe no significant difference in levels between MTAP intact and low MTAP expressing tumors, though MTA levels are generally higher in the lines with lower MTAP expression. They demonstrate that MTAP staining is intact in MTAP-WT and stromal tissue in human samples while absent in MTAP-deleted tumors. This is also true in xenografts. Having demonstrated that in vivo MTAP is present in the tissues surrounding the cell, the authors show by co-culture of MTAP-deleted and MTAP-intact cell lines that elevated MTA in the media of MTAP-deleted lines can be cleared from the media when MTAP-WT cells are present. As an indirect measure of PRMT5 activity (which is inhibited by elevated MTA), PRMT5 activity increases, leading to increased levels of SDMA, with co-culture leading to MTA clearance from the media. Overall, this is an interesting study but there are a couple critical issues that need to be addressed to make the data more convincing. Specifically, the authors need to strengthen data surrounding the lack of correlation between MTAP and MTA in vivo. MTA elevations have previously been reported in MTAP-deleted tumors and I am not convinced by how they defined MTAP-deleted in their GBM sample population.

- ⇒ Our study concerns MTA levels in primary human tumors of defined *MTAP*-deletion status, and we are not aware of any previous studies that have reported on this (No data on MTA levels of primary human *MTAP*-deleted vs. intact in major publications from Agios¹⁰, Boehringer Ingelheim⁹, and the Broad¹¹; even in Agio's more recent publication on *MTAP*¹², no MTA measurements were reported in primary human tumors). We searched the literature far and wide but did not find any prior studies that interrogated MTA elevations in *MTAP*-deleted vs. intact primary human tumors (we mean actual human tumors, not PDX in mice). If the reviewer knows

of studies of primary human tumors of defined *MTAP* genotype demonstrating increased MTA, we kindly ask that they share the reference with us.

⇒ We are only aware of two studies that have measured MTA in human tumors, neither of which determined *MTAP*-deletion status or attempted to ask whether *MTAP*-deletion resulted in MTA elevation. Kirovski *et al.*, 2011¹³ looked at the correlation between MTA levels in 19 human primary resected Hepatocellular Carcinoma (HCC) tumor samples. However, no *MTAP* genomic CNV status of individual tumors was presented in this study. Thus, it is not clear which HCC tumors were homozygous *MTAP*-deleted. In fact, the homozygous *CDKN2A/MTAP* deletion is quite rare in HCC (**The Below Figure From TCGA**¹⁴). The key difference between looking at the correlation between *MTAP* mRNA and MTA levels for HCC tumors versus GBM tumors (as we did in **Supplementary Figure S6**) is their frequencies of homozygous *MTAP* deletion. The homozygous *MTAP* deletion in HCC is rare (<3%¹⁵), while homozygous *MTAP* deletion is common in GBM (can reach 50% of cases^{6,16,17}). Please see **The Below Figure** obtained from cBioPortal¹⁴ comparing the homozygous *MTAP* deletion frequencies in HCC vs. GBM. Considering that less than 3% of HCC tumors are *MTAP* deleted, statistical chance would predict that none of the 20 tumors surveyed by Kirovski *et al.*¹³ would be homozygous *MTAP*-deleted.

***MTAP* homozygous deletion frequencies in GBM, HCC, and Melanoma**

GBM

Deep Deletion of *MTAP*: 42%

HCC

Deep Deletion of *MTAP*: 3%

Melanoma

Deep Deletion of *MTAP*: 18%

Frequencies of *MTAP* homozygous deletion across cancer types from TCGA Pan Cancer Atlas Studies.

- ⇒ Another study regarding measuring MTA levels in human tumors done by Stevens *et al.*¹⁸ in which authors measured MTA levels in a small number of primary human melanoma tumors (n=5). However, Stevens *et al.*¹⁸ did not attempt to determine whether MTA levels were higher in *MTAP*-deleted versus *MTAP*-intact primary melanoma tumors. This clarification is critical given that *MTAP* homozygous deletion only occurs in ~18% of melanomas, and the sample size is very small (n=5).
- ⇒ We should emphasize that our conclusion of “no significant elevation of MTA in homozygous *MTAP*-deleted human GBM tumors” (data are shown in **Figure 2** and **Supplementary Figure S5**) is based on tumors with verified homozygous *MTAP* deletion. The *MTAP* status of these tumors was determined from previous cGH studies (CNV data in Wang *et al.*⁸ and Kim *et al.*⁷), and the absence of *MTAP* in tumor cells verified by *MTAP*-IHC, with an antibody we validated extensively using *MTAP*-deleted xenografts (**Supplementary Figure S12-S14**).
- ⇒ Moreover, if the reviewer is thinking of a study that might have compared xenografted *MTAP*-deleted vs. WT tumors (rather than primary human tumors), we would not be surprised to find MTA elevations since xenografts have dramatically lower stromal infiltration than primary human GBM (**Supplementary Figure S15A vs. S15B-D**). Nevertheless, we are unaware of any published studies that have specifically compared MTA levels in *MTAP*-deleted versus intact PDX; we would again most appreciate if the reviewer would share a reference to such work that he is aware of.

Theoretically, *MTAP*-deleted tumors should exhibit no *MTAP* expression by western blot,

- ⇒ Short answer: Not true, since human GBM tumors are made up of a mix of malignant glioma cells with genetic alterations like homozygous *MTAP*-deletion, as well as non-transformed non-malignant stromal cells that do not carry any genetic abnormalities. Thus, even in homozygous *MTAP*-deleted tumors, only malignant glioma cells lack *MTAP*, while stromal cells express normal levels.

Figure from Behnan *et al.*, Brain, 2019. GBM tumors consist of malignant, transformed, and mutated glioma cells and non-malignant non-mutated stromal cells. Glioma cells in homozygous *MTAP*-deleted GBM tumors do not express *MTAP*. However, stroma cells are *MTAP*-WT and express *MTAP*. As a result, when a bulk resected GBM tumor is probed by Western blot and mRNA expression, *MTAP* protein is still expressed even if the glioma cells are *MTAP*-deleted and non-expressing.

⇒ Long answer: The supposition would theoretically true if a tumor were made up exclusively of cancer cells. However, this is not the case, with GBM tumors being made up of between 25-75% non-malignant stromal cells³⁻⁵. Thus, a homozygous *MTAP*-deleted GBM tumor is an admixture of non-malignant stromal cells (25-75%^{4,5}) that express *MTAP* and malignant glioma cells which do not express any levels of *MTAP*. Since a western blot is performed on lysates of the whole tumor, there is no reason to expect zero expression of *MTAP* due to the contribution of the non-malignant stromal proportion of the tumor to *MTAP* protein. However, on average, the homozygous *MTAP*-deleted tumors express lower *MTAP* protein levels by the western blot (please see the below graph). However, some expression of *MTAP* deriving from *MTAP*-WT

Figure 2H: *MTAP* protein levels corrected for GAPDH loading control from the western blot of HF series tumors shown in **Figure 2G**. Due to the presence of *MTAP*-WT cells in the lysate of whole tumor, *MTAP* protein levels are not zero for homozygous *MTAP*-deleted tumors. However, on average homozygous *MTAP*-deleted tumors have lower *MTAP* protein levels compared to intact tumors.

stromal cells will always be there. To illustrate microglia cells' presence in the lysate of whole tumors, we measured IBA1 protein levels by western blot (**Figure 2G** and **Supplementary Figure S5**). All tumors show some levels of IBA1, verifying myeloid stromal infiltration. In **Figure 2G**, note the homozygous *MTAP*-deleted HF3174 tumor has the lowest IBA1 compared to other homozygous *MTAP*-deleted tumors (indicating least stroma-containing tumor of this dataset), which also expresses the lowest but non-zero levels of *MTAP* (red arrows in **Figure 2G**).

The non-zero levels of *MTAP* protein for homozygous *MTAP*-deleted HF3174 tumor (marked with red arrow) is more evident in the higher exposed band of *MTAP*. In contrast, no *MTAP* expression is detected in the lysate of *MTAP*-deleted cells (U87).

Figure 2G: *MTAP* protein levels in bulk tumors are lower in *MTAP*-deleted compared to *MTAP*-intact ones but are not zero due to the presence of *MTAP*-expressing non-malignant stromal cells (e.g., microglia, astrocytes, and endothelial cells). The presence of microglia in a tumor is confirmed by the microglial markers IBA1 in tumor lysates but not seen in cells in culture.

⇒ The expression of MTAP in myeloid (macrophage/microglia) stroma in homozygous *MTAP*-deleted tumors is also validated in IHC of human GBM tumors. **Supplementary Figure S16** shows the strong correlation between residual MTAP staining and myeloid content (IBA1-positive cells) in homozygous *MTAP*-deleted human GBM tumors. In this figure, FFPE slides from *MTAP*-deleted tumors were stained with an anti-MTAP antibody with the *MTAP*-deleted glioma cells showing no staining, while non-malignant stromal cells, which are *MTAP*-WT such as microglia, intensely stained (black color). The staining pattern for IBA1 staining across different cases shows a strong correlation with MTAP staining in the *MTAP*-deleted tumors. It attests to the significant content of microglia/macrophages inside GBM tumors. This accounts for the differences in residual MTAP-expression in *MTAP*-deleted tumors when probed at the bulk tumor level (western blots). We should also note that myeloid cells are not the only *MTAP*-WT stromal cells present in GBM tumors. GBM tumors contain other *MTAP*-WT non-malignant cells such as glial, endothelial, and stem cells, contributing to the MTAP protein levels measured by Western blot.

Supplementary Figure S16

Supplementary Figure S16: Myeloid cells constitute the majority of *MTAP*-expressing stromal cells in *MTAP*-deleted GBM tumors. FFPE serial sections were stained against MTAP (first column) and IBA1 (myeloid marker, stains macrophages/microglia, second column) and images acquired from consecutive sections of the same tumor. (A) Case 503710, *MTAP*-deleted; (B) Case 997730, *MTAP*-deleted; (C) Case 506330, *MTAP*-deleted. Human GBM tumors may have up to 75% non-malignant stromal cell content, including *MTAP*-intact microglia. The presence of the *MTAP*-expression myeloid cells results in *MTAP*-positive stains even in GBM tumors that are homozygous *MTAP*-deleted, since only the malignant glioma cells carry the deletion.

⇒ To further validate that IBA1-positive myeloid cells drive the non-zero MTAP expression in bulk *MTAP*-deleted GBM tumors, we investigate the correlation between MTAP mRNA and AIF1 (the official gene symbol of IBA1, microglia/macrophages marker) mRNA levels among GBM tumors in an independent data set (TCGA⁶). **Supplementary Figure S16D** shows that there is a significant positive correlation between mRNA levels of MTAP and AIF1 in *MTAP*-deleted tumors vs. intact ones (but no positive correlation between MTAP and AIF in *MTAP*-intact tumors, fully consistent with the explanation, as in *MTAP*-intact tumors, both glioma and stromal cells express MTAP). This figure also indicates that homozygous *MTAP*-deleted GBM tumors, on average, have lower but non-zero levels of MTAP mRNA levels compared to *MTAP*-intact tumors, supporting our western blot data in **Figure 2G**.

Supplementary Figure S16D

Supplementary Figure S16D: The correlation between MTAP mRNA levels and AIF1 (microglia/macrophages marker) mRNA levels for homozygous *MTAP*-deleted (blue) and *MTAP*-intact (gray) human GBM tumors. Data obtained from TCGA cell 2016. GBM tumors are an admixture of cancer cells and non-stroma cells. Due to the presence of *MTAP*-expressing stroma cells such as microglia, homozygous *MTAP*-deleted GBM tumors express non-zero levels of MTAP. On average, homozygous *MTAP*-deleted tumors have lower MTAP mRNA levels than *MTAP*-intact and a stronger correlation between IBA1 and MTAP for homozygous *MTAP*-deleted GBM tumors.

⇒ To summarize: the non-zero MTAP expression measured on lysate of bulk homozygous *MTAP*-deleted GBM tumors is derived from non-malignant *MTAP*-intact stromal cells and not malignant glioma cells. Malignant glioma cells in homozygous *MTAP*-deleted tumors do not express any MTAP protein.

and there are certainly patient-derived GBM glioma models in use that do not have detectable protein levels of MTAP.

⇒ While we are unaware of a specific paper that includes a western blot of *MTAP*-deleted vs. *MTAP*-WT PDX's, we would not be surprised by such a result for the simple reason that xenografts have dramatically lower *MTAP*-expressing stroma (non-malignant cells) content than primary human tumors. To illustrate this, we present immunohistochemical *MTAP*-staining data from *MTAP*-deleted human xenografted tumors in mice versus *MTAP*-deleted primary human GBMs. The dramatic difference in residual *MTAP* expression in stroma is illustrated nicely by comparing the IHC for *MTAP* in U87 (homozygous *MTAP*-deleted **Supplementary Figure S15A**) intracranial xenografts vs. homozygous *MTAP*-deleted primary GBM tumors (**Supplementary Figure S15B-D**).

Supplementary Figure S15

Supplementary Figure S15: Lower stromal content in human xenograft in mice compared to human primary GBM tumors. Tumors are stained against MTAP antibody and images acquired using a 40X objective. (A) *MTAP*-deleted intracranial xenografted in mice (U87), (B-D) *MTAP*-deleted human GBM tumors with varying degrees of stromal content (minimal to extreme stromal content). The degree of *MTAP*-positive stromal infiltration was much more significant in the primary human GBM tumors than xenografts.

⇒ While it is already well established that primary human GBMs have a much higher proportion of stromal components such as astrocytes and myeloid cells than xenografts, this concept is further illustrated in the **Supplementary Figure S15E-H** comparing the myeloid stroma levels of homozygous *MTAP*-deleted human xenograft (E, Gli56) vs. primary human GBM tumors (F-H).

Supplementary Figure S15E-H

Supplementary Figure S15E-H: Negligible stromal infiltration in human xenografts in mice compared to primary GBM tumors. FFPE sections were stained against IBA1 (myeloid marker, black). (E) *MTAP*-deleted intracranial xenografted in mice (Gli56), (F-H) *MTAP*-deleted human GBM tumors with varying degrees of IBA1-positive cells (stromal content). The amount of IBA1-positive (myeloid) cells are dramatically higher in human GBM tumors (even in an example of “low” stromal content) than xenografts.

⇒ Also, there is minimal IHC staining for MTAP in *MTAP*-deleted U87 and Gli56 xenografts vs. strong MTAP staining in the normal mouse brain (**Supplementary Figure S12-S14**), confirming negligible stromal infiltration in human xenografted tumors.

Supplementary Figure S13

Supplementary Figure S14

Panels from **Supplementary Figure S12** and **S14** confirming negligible stromal infiltration in *MTAP*-deleted human xenografts (U87 and Gli56) compared to normal mouse brain. Tumor boundaries are shown in yellow. Note the intense staining pretty much in all cells except those with *MTAP*-deletions. This fully validates that this genuinely detects *MTAP* protein by IHC in FFPE sections.

Here, however, the authors are using *MTAP*-deletion and low *MTAP* interchangeably but they have not provided sufficient evidence that this is appropriate. This makes me wonder if a significant correlation between *MTAP* loss and MTA elevation might be found if their definition/threshold for “low *MTAP* expression” was more stringent. *MTAP*-deletion and “low *MTAP* expression” are not necessarily the same and I worry that the authors assumption of this leads to inaccurate conclusions.

⇒ First, please note that we did not use the “*MTAP*-deletion” and “low *MTAP* expression” interchangeably. Our conclusion of “insignificant elevation of MTA in homozygous *MTAP*-deleted GBM tumors” has derived from comparing MTA levels between homozygous *MTAP*-deleted tumors and *MTAP*-intact, as defined by genomic array cGH studies performed in reference⁷. Malignant glioma cells in homozygous *MTAP*-deleted tumors do not express any *MTAP* expression (verified by IHC); *MTAP* expression is only evident in the non-malignant stroma of such tumors. Heterozygous *MTAP*-deleted and *MTAP*-WT tumors in which cancer cells inside tumors express some levels of *MTAP* are considered as *MTAP*-intact for our purposes since only homozygous *MTAP*-deleted cells accumulate MTA in culture.

⇒ In the first submission of the paper, we first discussed MTA levels in GBM tumors from public domain data, and then we discussed our metabolomic profiling data. For the public domain data (Previously **Supplementary Figure S2** and now **Supplementary Figure S6**), we do not have information on *MTAP* deletion status, and we do not claim to know *MTAP* status for each tumor. In the new version of the paper, we first discuss our profiling data (MTA levels in GBM tumors with verified *MTAP*-deletion status, **Figure 2** and **Supplementary Figure S5**). We then used the public domain data to support our argument as independent measurements. We hope changing the flow of the paper resolves this confusion. Please see the reply to the following comment regarding clarification of previously **Supplementary Figure S2** and now **Supplementary Figure S6**.

⇒ The *MTAP*-deletion status of GBM tumors (HF series) in **Figure 2** was determined by array comparative genomic hybridization (array cGH) studies^{7,8} and further verified by immunohistochemistry. The genomic copy number variation of HF series tumors (**Figure 2B**) was profiled previously⁷, and the *MTAP* deletion data was obtained from this publication’s publicly deposited data. Briefly, DNA was isolated from whole tumors and hybridized with SNP 6.0 affi arrays. Copy numbers were corrected for tumor cellularity and whole-genome duplications. Our assignment of homozygous *MTAP* deletion versus *MTAP*-WT status for the HF series’ tumors agrees with the *CDKN2A* status of the tumors assigned by experts in the genomic field in Kim *et al.*⁷ paper. Please

Figure2B: Genomic copy-number data (dark blue: homozygous deletion, dark red: amplification) around the 9p21 locus. Each strip in the y-axis represents a single tumor at the specific chromosomal location (x-axis). Tumors with heterozygous *MTAP*-deletion and *MTAP*-WT tumors are categorized as *MTAP*-intact in our manuscript.

CDKN2A status of the tumors assigned by experts in the genomic field in Kim *et al.*⁷ paper. Please

see below the table from Kim *et al.*⁷ indicating the *CDKN2A* status of HF tumors. Note that about 70-80% of *CDKN2A* homozygous deleted GBM tumors are *MTAP* homozygous co-deleted.

**Figure 3A Kim *et al.*, Genome Res 2015
(The assignment of *CDKN2A* deletion status)**

Assignment of *CDKN2A* deletion status of individual tumors in the HF series by Kim *et al.* Genome Res 2015. The co-deletion of *MTAP* may be observed in 80%-90% of all tumors harboring the homozygous deletion of *CDKN2A*⁶. Except for HF2898 and HF3041, all homozygous *CDKN2A*-deleted tumors harbor homozygous *MTAP*-deletion as well. Tumors with heterozygous *MTAP*-deletion and *MTAP*-WT tumors are categorized as *MTAP*-intact in our manuscript.

⇒ Moreover, we should emphasize that MTA elevation and metabolic vulnerabilities caused by it are only evident in *MTAP*-homozygous deleted cells in culture and not heterozygous (mono-allelic deleted; low expressing) or WT (**The Below Figure**, data re-plotted from Kryukov *et al.*⁹). As such, in our study, we grouped *MTAP* homozygous deleted alone versus all others.

MTA accumulates in homozygous *MTAP*-deleted cells in culture and not heterozygous *MTAP*-deleted or WT cell lines. Data are re-plotted from the supplementary information of Kryukov *et al.* MTA levels are significantly higher in homozygous *MTAP*-deleted cells compared to heterozygous *MTAP*-deleted and *MTAP*-WT cells in culture. There is no statistical difference between intracellular levels of MTA between heterozygous *MTAP*-deleted and *MTAP*-WT cells in culture. %%% $P < 0.05$, *** $P < 0.05$.

⇒ In the revised version of this paper, we have performed and included additional metabolomic profiling data (**Supplementary Figure S5**). Instead of the BIDMC platform that we used to measure MTA levels for the HF series, we used Metabolon, Inc. platform as the independent verification. The *MTAP*-deletion status of tumors in the MDA series (**Supplementary Figure S5**) was confirmed by immunohistochemistry against the validated *MTAP* antibody.

Supplementary Figure S5

Supplementary Figure S5: No significant elevation of MTA in *MTAP*-deleted primary resected GBM tumors in an independent metabolomic data set. (A) Flash-frozen primary resected GBM tumors (heterogeneous mix of transformed glioma cells and non-malignant stromal cells) were prepared for IHC, mass spectrometry, and Western blotting. (B) *MTAP* deletion status of tumors was confirmed by immunohistochemistry using a validated *MTAP* antibody. Representative cases of *MTAP* positive and *MTAP* negative GBM tumors are shown. Non-malignant *MTAP*-intact cells are stained positive in the *MTAP*-deleted case. (C) MTA levels (absolute ion counts) in *MTAP*-deleted (blue) versus *MTAP*-intact (gray) GBM tumors, using the Metabolon, Inc. platform. Each bar represents MTA levels for each tumor. There is a 1.14-fold increase in the median MTA levels in *MTAP*-deleted tumors compared to intact tumors ($p = 0.9$, unpaired 2-tailed t-test with unequal variance). (D, E) Same data but expressed as a ratio of each tumor's total ion count for sample loading normalization (D) and as a ratio to SAM levels to account for upregulation in the methionine salvage pathway for each tumor (E). The median MTA/sum and MTA/SAM fold change in *MTAP*-deleted tumors compared to intact tumors are 1.23 ($p=0.9$) and 0.8 ($p=0.3$) respectively. (F) The levels of vinculin, *MTAP*, and IBA1 (microglia marker) in some of the tumors.

How exactly was MTAP expression measured in Supplementary Figure 2 and what is the label of the y-axis. Is this protein level (western blot/IHC), mRNA, something else? mRNA level is not always equivalent to protein abundance. A tumor with homozygous deletion of 9p21 would be CN = 0 and therefore no expression at all. Along these lines, MTAP expression is observed in the tumors that the authors are calling “MTAP deleted”. Low (but present) MTAP protein level may be sufficient to regulate MTA.

⇒ This point goes to the same principle elaborated on in the previous reply: When assayed at the bulk tumor level, MTAP-deleted tumors still express MTAP because of non-malignant, MTAP-expressing stromal cells. So, we agree with the reviewer that on its own, when measured in bulk tumors, low MTAP expression does not prove homozygous *MTAP*-deletion, yet, *MTAP*-deleted tumors, as defined by genomic copy number, show on average, lower expression of MTAP even when assayed in bulk; this is evident both in our western blot data as well as in RNA seq data from TCGA¹⁴ (**Supplementary Figure S6G**). For data shown in the initial manuscript in **Supplementary Figure S2** (public domain MTAP mRNA with MTA metabolite levels, Prabhu *et al.*¹⁹) and now in the revision shown in **Supplementary Figure S6A-F**, we have sorted the tumors based on their MTAP mRNA level (y-axis, Affy units). However, we did not claim and categorize individual tumors as homozygous *MTAP*-deleted and *MTAP*-intact. The overall frequency of *MTAP* homozygous deletion in GBM tumors is about 50%. That said, in the population of 50 tumors, it is much more likely that tumors with low bulk MTAP expression belong to the 50% of the group with homozygous *MTAP*-deletion vs. 50% *MTAP*-intact. To illustrate this point, consider the expression of MTAP measured by RNA seq in TCGA with GBM tumors of defined *MTAP*-deletion status, sorted by quartiles of MTAP mRNA expression (**Supplementary Figure S6G**). Note that the vast majority (though clearly, not all) of the *MTAP*-genomically deleted (blue dots) tumors fall in first and second quartiles of low MTAP expression, while the bulk of *MTAP*-intact tumors fall into the third and fourth quartiles of higher mRNA MTAP expression. Based on this, it is reasonable to infer that in the **Supplementary Figure S6A-F** (Prabhu *et al.*¹⁹), if tumors are ranked by quartiles of MTAP expression, the vast majority of *MTAP*-deleted cases will be found in first and second quartiles. Data on **Supplementary Figure S6** are provided as an additional example to support our conclusion from tumors with known *MTAP*-deletion status (**Figure 2** and **Supplementary Figure S5**).

Supplementary Figure S6

Supplementary Figure S6: No correlation between MTAP expression and MTA levels in primary human GBM tumors in an independent dataset. (A-F) Data replotted from Prabhu *et al.*, *Neuro. Oncol.*, 2019¹⁰, where metabolomic profiling was performed by Metabolon Inc. and gene expression analysis was performed by Affymetrix Human expression array. (A) MTAP mRNA levels of 50 primary human GBM tumors sorted from low to high, (B) MTA levels normalized to the median, and (C) MTA levels as a ratio of total ion count to account for differential loading, sorted based on their MTAP expression in A. (D) The scatter plot of MTA levels versus MTAP expression with an insignificant coefficient of correlation highlighting no matter how we analyze these data, the results came insignificant. (E) MTA levels (mean + SD) among the same 50 human primary GBM tumors sorted based on MTAP expression (A, F) and divided into four quartiles. The frequency of *MTAP* deletion in GBM tumors is about 50%. That said, in the population of 50 tumors, it is much more likely that tumors with low bulk MTAP expression belong to the 50% of the group with homozygous *MTAP*-deletion vs. 50% *MTAP*-intact. In other words, the vast majority of homozygous *MTAP*-deleted tumors fall in the first and second quartiles of low MTAP expression. In contrast, the bulk of *MTAP*-intact tumors fall into the third and fourth quartiles of higher mRNA MTAP expression. Comparing MTA levels between quartiles suggests no significant elevation of MTA in quartiles with low MTAP expression vs. high expression (mean MTA levels are 1.20, 1.50, 1.15, and 1.27 for each quartile, $p = 0.48$, Anova: single factor). This figure supports our conclusion from tumors with known *MTAP*-deletion status (Figure 2 and Supplementary Figure S5). (G) *MTAP* mRNA levels of 147 GBM tumors with known *MTAP*-deletion status from the TCGA dataset (Firehose Legacy¹¹) to illustrate that majority (not all) of homozygous *MTAP*-deleted tumors fall in the first and second quartiles of low MTAP expression. *MTAP* mRNA expression was sorted from low to high and divided into four quartiles. Since the frequency of *MTAP* deletion in GBM is about 50%, most tumors in the two lowest quartiles of *MTAP* mRNA expression are homozygous *MTAP*-deleted. In contrast, the majority of the tumors in the last two quartiles are *MTAP*-intact.

In Figure 3, it would be helpful to see how MTA levels correlate with amount of MTAP protein present by western blot.

- ⇒ Please see the below graph, which shows the MTA levels versus the amount of protein levels present by Western blot. No significant correlation was found.

Data in Supplementary Figure 4 do not support your other data – these show that MTA levels (normalized to mean or normalized to SAM) are significantly higher in MTAP-deleted tumors. Why the discrepancy? And, are these tumor truly MTAP-deleted or MTAP-low?

- ⇒ Please note that Supplementary Figure S4 illustrates MTA and SAM levels in cells in culture and not human GBM tumors. We re-wrote the figure legend to emphasize that these data represent metabolites levels of cells in culture, not human tumors. We are all in agreement, *for cells in culture without stromal cells*, MTA levels are higher in MTAP-deleted cancer cell lines than WT. These data, which were re-plotted from the supplementary information of different published studies^{9,20}, indicate that SAM levels are not different between MTAP-deleted and wild-type cancer cell lines in culture. These data justify using SAM as an intracellular metabolite for the normalization of MTA.

Supplementary Figure 6D and its conclusions are based on assumptions and not known genotypes of the patients being sampled. This is a weak argument.

- ⇒ Supplementary Figure S6 is now shown as Supplementary Figure S11. We agreed that for this figure, knowing the MTAP deletion status of tumors will strengthen the argument. Unfortunately, such data were not collected and are not available. However, we want to point out that the argument is based on comparing a group with a high incidence of homozygous MTAP-deletion (grade IV GBM ~50% MTAP-deleted) with lower or zero incidence of homozygous MTAP-deletion (low-grade glioma and normal brain). Homozygous MTAP deletion frequency can reach 50%⁶ in GBM tumors, while its incidence is about 10%²⁰ in low-grade glioma and zero in a normal brain. Using the binomial distribution, the chance of having at least one MTAP-deleted tumor (among 10 tumors) for Supplementary Figure S11A is 99.9%. The probability of having at least one MTAP-deleted tumor- among 6 tumors- for Supplementary Figure S11C-S11E is 98.4%. However, the probability of having MTAP-deleted samples for the normal brain (Supplementary Figure S11A) and low-grade glioma (Supplementary Figure S11C-S11E) is zero and 46%, respectively. In our argument in the manuscript, we clearly state that the argument is based on likelihood (assumption) and not knowing the patients' genotypes.

Figure 1A is not data but summary of the pathway – perhaps the TCGA data should be omitted since this is not primary data.

- ⇒ We moved TCGA data to **Supplementary Figure S1**. TCGA data are public domain data free to use in publications and critical to supporting genomic studies.

It is difficult to read the y-axis units on Figure 2C – is this 10Z?

- ⇒ Please note that the y-axis on this figure is 10^7 . We re-plotted the figure to make the y-axis to be more apparent.

Please be cautious in the discussion of MAT2A as a target. Though MAT2A's important cellular role is noted, many targeted therapies are aimed at essential genes (ex EGFR null mice have early lethality) – it is the therapeutic window that is attempted to be exploited with MAT2A inhibition in the setting of MTAP deletion. For mechanism related to other inhibitors it may prove useful to others.

- ⇒ We fully agree with the reviewer that MAT2A inhibition may very well prove efficacious for cancer treatment, as this is an essential gene. However, our argument centers on whether a therapeutic window for MAT2A and PRMT5 inhibitors generated by homozygous *MTAP*-deletion is translatable to human primary GBM tumors. To address the reviewer's comment and to clearly state that our results do not detract from other avenues of utilizing MAT2A inhibitors in cancer treatment, we have modified our discussion as below:

“That *MTAP* deletions can be leveraged as a point of selective vulnerability in various cancers has spurred much excitement. Our analysis challenges whether the metabolic conditions required for therapies to exploit vulnerabilities associated with elevated MTA/ *MTAP*-deletions are present in primary human tumors, giving pause to whether these would translate to the clinic. Yet, our findings do not rule out that MAT2A and or PRMT5 inhibitors could prove useful in oncology. Merely, such inhibitors would have to achieve a therapeutic window through a mechanism other than MTA accumulation. Indeed, recent research has already pointed towards *MTAP*-deletion independent sensitization mechanism to PRMT5 inhibition⁴⁷. Some GBM tumors in our dataset showed quite low levels of SDMA levels —suggesting low PRMT5 activity— without *MTAP*-deletions (**Figure 2**). Perhaps such tumors with low PRMT5 activity could be suitable candidates for targeted therapies against PRMT5 or MAT2A, provided a reliable molecular marker can be found to identify them.”

The GBM tumors label is out of place on Suppl. Fig 3.

- ⇒ Thanks for pointing it out; we have fixed this issue.

Reviewer #3 (Remarks to the Author); expert on MTAP-deleted targeting:

A major conclusion from this paper is that MTA is excreted by GMB cells and is metabolized by stromal cells. Evidence for products of metabolism of MTAP would further strengthen their conclusions.

⇒ To address the concerns raised by the reviewer, we performed a series of experiments to 1) determine if the release of functional MTAP enzyme by *MTAP*-WT cells into the media is responsible for eliminating exogenous MTA from conditioned media and 2) isotope labeling studies to demonstrate that exogenous MTA is taken up by *MTAP*-intact cells and converted to methionine, via MTAP and the methionine salvage pathway.

⇒ First, we sought to see if the release of functional MTAP enzyme by *MTAP*-WT cells is responsible for eliminating exogenous MTA from conditioned media. We grew *MTAP*-WT RAW-264.7 (macrophages) cells in DMEM; after 3 days, we collected the conditioned media and spun it down to remove any cells, then we incubated the conditioned media collected from RAW-264.7 cells with 20 μ M exogenous MTA for 3 more days. **Supplementary Figure S19A** shows that incubation of the conditioned media collected from RAW-264.7 cells with exogenous MTA does

Supplementary Figure S19A

Supplementary Figure S19A: *MTAP*-intact cells (macrophages) were cultured with or without exogenous MTA (20 μ M vs. 0 μ M). Exogenous MTA was not eliminated from the cell-free macrophage conditioned media, while it disappeared from media cultured with macrophages. This result indicates that the release of functional MTAP enzyme by the macrophages to the extracellular environment does not contribute to exogenous MTA's elimination by *MTAP*-WT cells. Direct evidence for exogenous MTA consumption and metabolism in *MTAP*-intact cells.

not result in abrogation of MTA. This result indicates that the release of functional MTAP enzymes does not contribute to the disappearance of exogenous MTA from the conditioned media of *MTAP*-WT cells.

⇒ Next, we sought to investigate if exogenous MTA is taken up and metabolized by *MTAP*-WT cells. Thus, we cultured *MTAP*-WT macrophages (RAW-264.7) or glioma cells (D423) with labeled methyl tri-deuterated-MTA (D3-MTA) (**Figure 4A**). We observed the rapid elimination of labeled D3-MTA from conditioned media of *MTAP*-WT followed by the deuterium-enrichment of intracellular MTA, methionine, and SAM (**Figure 4B, C, and Supplementary Figure S19C, D**). **Figure 4D** shows the deuterium's fate from the labeled D3-MTA into methionine (methionine salvage pathway) and SAM (polyamine biosynthesis). This result indicates that extracellular MTA can be taken up and metabolized by *MTAP*-WT cells.

Figure 4: Exogenous MTA is consumed by *MTAP*-intact cells through the methionine salvage pathway. (A) *MTAP*-WT macrophages (RAW-264.7) were cultured with 100 μ M methyl tri-deuterated-MTA (D3-MTA). Cells and conditioned media were extracted and prepared for LC-MS and NMR measurements, respectively. (B) Exogenous D3-MTA rapidly disappears from the media in cultures of macrophages, as the deuterium label (Mass +3; M+3) appears in intracellular methionine and SAM (C), consistent with *MTAP*-dependent metabolism by the methionine salvage pathway (D).

Supplementary Figure S19

Supplementary Figure S19: (C) Culturing *MTAP*-WT (glioma) cells with exogenous deuterated MTA results in the abrogation of D3-MTA from the conditioned media. **(D)** Intracellular enrichment of MTA, methionine, and SAM after culturing *MTAP*-WT glioma cells with the (M + 3, D3) D3-MTA label in methionine and SAM. This data indicates that MTA is taken up by *MTAP*-intact cells and metabolized further, all the way to methionine through the methionine salvage pathway.

References:

1. Sanderson, S. M., Mikhael, P. G., Ramesh, V., Dai, Z. & Locasale, J. W. Nutrient availability shapes methionine metabolism in p16/ *MTAP* -deleted cells. *Sci. Adv.* **5**, eaav7769 (2019).
2. Shlomi, T., Fan, J., Tang, B., Kruger, W. D. & Rabinowitz, J. D. Quantitation of cellular metabolic fluxes of methionine. *Anal. Chem.* **86**, 1583–1591 (2014).
3. Behnan, J., Finocchiaro, G. & Hanna, G. The landscape of the mesenchymal signature in brain tumours. *Brain* **142**, 847–866 (2019).
4. Charles, N. A., Holland, E. C., Gilbertson, R., Glass, R. & Kettenmann, H. The brain tumor microenvironment. *Glia* **59**, 1169–1180 (2011).
5. Darmanis, S. *et al.* Single-Cell RNA-Seq Analysis of Infiltrating Neoplastic Cells at the Migrating Front of Human Glioblastoma. *Cell Rep.* **21**, 1399–1410 (2017).
6. Brennan, C. W. *et al.* The Somatic Genomic Landscape of Glioblastoma. *Cell* **155**, 462–477 (2013).
7. Kim, H. *et al.* Whole-genome and multisector exome sequencing of primary and post-treatment glioblastoma reveals patterns of tumor evolution. *Genome Res.* **25**, 316–327 (2015).
8. Wang, Q. *et al.* Tumor Evolution of Glioma-Intrinsic Gene Expression Subtypes Associates with Immunological Changes in the Microenvironment. *Cancer Cell* **32**, 42–56.e6 (2017).
9. Kryukov, G. V. *et al.* *MTAP* deletion confers enhanced dependency on the PRMT5 arginine methyltransferase in cancer cells. *Science* (80-.). **351**, 1214–1218 (2016).
10. Marjon, K. *et al.* *MTAP* Deletions in Cancer Create Vulnerability to Targeting of the MAT2A/PRMT5/RIOK1 Axis. *Cell Rep.* **15**, 574–587 (2016).
11. Mavrakis, K. J. *et al.* Disordered methionine metabolism in *MTAP/CDKN2A*-deleted cancers leads to dependence on PRMT5. *Science* (80-.). **351**, 1208–1213 (2016).
12. Kalev, P. *et al.* MAT2A Inhibition Blocks the Growth of *MTAP*-Deleted Cancer Cells by Reducing PRMT5-Dependent mRNA Splicing and Inducing DNA Damage. *Cancer Cell* **39**, 209–224.e11 (2021).
13. Kirovski, G. *et al.* Down-Regulation of Methylthioadenosine Phosphorylase (*MTAP*) Induces Progression of Hepatocellular Carcinoma via Accumulation of 5'-Deoxy-5'-Methylthioadenosine (*MTA*). *Am. J. Pathol.* **178**, 1145–1152 (2011).
14. Cerami, E. *et al.* The cBio Cancer Genomics Portal: An open platform for exploring multidimensional cancer genomics data. *Cancer Discov.* **2**, 401–404 (2012).
15. Ally, A. *et al.* Comprehensive and Integrative Genomic Characterization of Hepatocellular Carcinoma. *Cell* **169**, 1327–1341.e23 (2017).
16. Chinnaiyan, P. *et al.* Molecular and Cellular Pathobiology The Metabolomic Signature of Malignant Glioma Reflects Accelerated Anabolic Metabolism. *Cancer Res.* **72**, 5878–88. (2012).
17. Elkhaled, A. *et al.* Characterization of metabolites in infiltrating gliomas using ex vivo ¹H high-resolution magic angle spinning spectroscopy. *NMR Biomed.* **27**, 578–593 (2014).
18. Stevens, A. P. *et al.* Direct and tumor microenvironment mediated influences of 5'-deoxy-5'-(methylthio)adenosine on tumor progression of malignant melanoma. *J. Cell. Biochem.* (2009) doi:10.1002/jcb.21984.
19. Prabhu, A. H. *et al.* Integrative cross-platform analyses identify enhanced heterotrophy as

- a metabolic hallmark in glioblastoma. *Neuro. Oncol.* **21**, 337–347 (2019).
20. Ortmayr, K., Dubuis, S. & Zampieri, M. Metabolic profiling of cancer cells reveals genome-wide crosstalk between transcriptional regulators and metabolism. *Nat. Commun.* **10**, 1841 (2019).
 21. Sachamitr, P. *et al.* PRMT5 inhibition disrupts splicing and stemness in glioblastoma. *Nat. Commun.* **12**, 1–17 (2021).

REVIEWER COMMENTS

Reviewer #1 (Remarks to the Author):

In their revised manuscript, the authors have done an extensive effort to address the referees' concern, and they should be commended for doing this work under these exceptional circumstances.

Some of the new experiments helped to address this referee's concerns. The conclusion that MTAP-negative cells secrete MTA and that macrophages can uptake it is convincing. There are still some remaining concerns:

The presentation of these data from Sanderson et al seem to underplay the effects of external cues on MTA accumulation. Indeed, the authors present the data upon Log10 transformation, which significantly reduces the dynamic range of the results (Fig S9). I advise presenting these results showing the peak area, as done for all the other experiments, unless justified appropriately. This result would be important since it may explain the lack of MTA accumulation in vivo, in GBM, is due to very low level of cysteine that the tumour experience.

Also, I still believe that the comparison between the intracellular and extracellular level of MTA should not be used to conclude that MTAP-negative cells accumulate "less MTA" than what they secrete (note that the absolute quantification proposed by the authors does not address this point). Indeed, considering that the accumulation of MTA in the media is strongly time-dependent (as the authors point out), this comparison still seems misleading, but I leave to the authors the decision to tone down the conclusions from this result.

The remaining concern is about the measurements of MTA in tumour tissues. As the authors now discuss and clarify, GBM specimens are highly heterogeneous and likely contain a substantial proportion of stromal cells. When performing metabolomics from these samples, it is therefore possible that the MTA signal from the MTAP-negative cancer cells is somewhat diluted by the MTAP-positive cells. This issue can explain many discrepancies indicated by the authors, including the lack of significant difference in MTA and methionine levels between MTAP-deleted and intact human GBM tumors. The same concern applies to the western blots, where even the SDME-Arg signal could be a mix of the level of this modification in cancer and stromal cells (as shown by MTAP level itself). The authors show that the HF3174 tumor present less stromal contamination and indeed this tumour shows a low SDME-Arg signal, arguing that MTA could be high in this tumour. Considering that the main message of the paper is based upon this experimental setting, the authors should make sure that the signal they measure in vivo is a genuine readout of the metabolite levels in cancer cells, and not a reflection of a dilution effect from the stroma. The referee is aware that this request is technically very challenging, so another possible option is to clarify the limitations of the in vivo studies and tone down the conclusions.

Reviewer #2 (Remarks to the Author):

The manuscript is improved. The author has rewritten to more precisely and accurately describe the results presented.

Reviewer #3 (Remarks to the Author):

Authors have responded well to queries of other reviewer.

REVIEWERS' COMMENTS

Reviewer #1 (Remarks to the Author):

In their revised manuscript, the authors have done an extensive effort to address the referees' concern, and they should be commended for doing this work under these exceptional circumstances.

⇒ We thank the reviewer for their attention and time in reviewing our manuscript, as well as acknowledging the difficulties under which we had to work.

Some of the new experiments helped to address this referee's concerns. The conclusion that MTAP-negative cells secrete MTA and that macrophages can uptake it is convincing. There are still some remaining concerns:

The presentation of these data from Sanderson et al seem to underplay the effects of external cues on MTA accumulation. Indeed, the authors present the data upon Log10 transformation, which significantly reduces the dynamic range of the results (Fig S9). I advise presenting these results showing the peak area, as done for all the other experiments, unless justified appropriately. This result would be important since it may explain the lack of MTA accumulation in vivo, in GBM, is due to very low level of cysteine that the tumour experience.

⇒ We have replotted the data in **Supplementary Figure S9** using linear axis and MS intensity (the same unit reported in Sanderson et al.¹). To be clear, the statistical test had been performed on linear, not log-transformed data, so the linear replotting does not fundamentally change the interpretation.

Also, I still believe that the comparison between the intracellular and extracellular level of MTA should not be used to conclude that MTAP-negative cells accumulate "less MTA" than what they secrete (note that the absolute quantification proposed by the authors does not address this point). Indeed, considering that the accumulation of MTA in the media is strongly time-dependent (as the authors point out), this comparison still seems misleading, but I leave to the authors the decision to tone down the conclusions from this result.

⇒ We acknowledge the point raised by the reviewer and accept that with present data, this is where reasonable people may disagree. The in vitro intracellular measurements of MTA, as noted, vary extensively across studies and appear to be influenced by the degree of pellet washing (**Supplementary Figure S2**). Therefore, the question of whether MTA levels are higher extracellularly or intracellularly depends on the technicalities of how the experiment is conducted and on metrics such as cytosolic volume (to calculate the molarity of MTA intracellularly).

⇒ Nevertheless, we believe the data strongly support our main contention (which we acknowledge is slightly different from the point the reviewer is making here) that the most significant difference (in terms of MTA) between *MTAP*-deleted and *MTAP*-intact cancer cells is how much MTA they secrete. Indeed, MTA levels in conditioned media are near zero for *MTAP*-intact cell lines, but in the uM range for *MTAP*-deleted ones. This finding is independently corroborated by retrospective analysis of public domain metabolomic data. These conditioned media measurements of MTA are the least technically challenging and most reliable.

The remaining concern is about the measurements of MTA in tumour tissues. As the authors now discuss and clarify, GBM specimens are highly heterogeneous and likely contain a substantial proportion of stromal cells. When performing metabolomics from these samples, it is therefore possible that the MTA signal from the *MTAP*-negative cancer cells is somewhat diluted by the *MTAP*-positive cells. This issue can explain many discrepancies indicated by the authors, including the lack of significant difference in MTA and methionine levels between *MTAP*-deleted and intact human GBM tumors.

⇒ Metabolomics profiles on bulk tumor have been standard in the field (and the lack of high sensitivity in vivo metabolic imaging techniques), but we acknowledge that not being able to differentiate the MTA signal from malignant and non-malignant cells is an inherent limitation. However, we would like to mention a few points in defense of using this method to determine whether an elevation of MTA in *MTAP*-deleted GBM tumors is evident or not:

1. Elevation of specific oncometabolites in heterogeneous GBM tumors can be readily evident in whole tumors' metabolomics profiling data – metabolomics performed on heterogeneous tumors containing both malignant glioma and stromal cells. The elevation of 2-hydroxyglutarate (2-HG) in *IDH1* mutant tumors is one example which is shown in **Figure 2f** and **Supplementary Figure S7**.

Figure 2f: tumors with *IDH1* mutation stand out by their dramatic elevation of 2-hydroxyglutarate (2-HG), providing a positive control for genomic/metabolic correlation.

- Given that the reported MTA levels of *MTAP*-deleted cells in culture can be elevated as high as 50-fold compared to *MTAP*-intact cells (**Below Figure^{2,3}**), it is reasonable to conclude that if this metabolic alteration (such high elevation) held true in vivo, it should also be seen in heterogeneous tumors as well (as elevation of 2-HG can be seen in IDH1 mutant GBM tumors). GBM tumor cellularity (proportion of cells in a

Figure 4A from Mavrakis et al. showing dramatically high levels of intracellular MTA in *MTAP*-deleted cells

Figure 3C from Marion et al. showing dramatically high levels of intracellular MTA in *MTAP*-deleted cells

Mavrakis et al. and Marion et al. reported on exploiting *MTAP*-deletion/MTA accumulation as the metabolic vulnerability, have also reported very high levels of intracellular MTA in *MTAP*-deleted cells than WT.

tumor that are malignant) is typically between 20% and 75%, on average 50%. With a 50% glioma and 50% stromal content, based on the MTA levels in *MTAP*-deleted vs. WT cells as reported by Marion et al.³ in Figure 3C (shown above), we would still expect a many-fold higher level of MTA in *MTAP*-deleted primary GBM tumors compared to intact ones (with the difference to be significant). That we do not see this, and that we do not see a significant decrease in SDMA, suggests that MTA levels do not recapitulate the increase seen in cells in culture.

- However, our data of profiling GBM tumors from two different data set (HF series and MDA series) shows that there are nonsignificant 1.4-fold ($p=0.2$) and 1.14-fold ($p=0.9$) increase in MTA levels in *MTAP*-deleted tumors vs. intact ones. While our data do not rule out small increases, these would have to be dramatically lower than those observed in culture.

The same concern applies to the western blots, where even the SDME-Arg signal could be a mix of the level of this modification in cancer and stromal cells (as shown by *MTAP* level itself). The authors show that the HF3174 tumor present less stromal contamination and indeed this tumour shows a low SDME-Arg signal, arguing that MTA could be high in this tumour.

1. **Figure 2G** shows that the primary tumors showed no meaningful decrease in levels of SDMA (SDMe-Arg), representing PRMT5 activity, in homozygous *MTAP*-deleted compared to *MTAP*-intact human GBM tumors. While the western blot was done on whole GBM tumors and the signal came from both cancer cells and microglia cells, there are two tumors with low but nonzero levels of IBA1 expression (myeloid marker): HF3174 (*MTAP*-deleted) and HF2990 (*MTAP*-intact), marked in the **Below Figure**. Comparing the SDMA levels of these two tumors with low IBA1 expression, we can see that the *MTAP*-deleted HF3174 tumor has slightly more expression of SDMA compared to the *MTAP*-intact HF2990 tumor. While the *MTAP*-deleted HF3174 tumor "shows low SDMe-Arg signal," as the reviewer mentioned, two *MTAP*-intact tumors (HF2990 and HF3050) have even lesser SDMA signal compared to the *MTAP*-deleted HF3174 tumor.

Figure 1g: SDMA levels measured in GBM tumors.

2. While the SDMA levels measured come from the mixed signal from cancer cells and stroma, there should still be a trend showing that, on average, *MTAP*-deleted tumors should express less SDMA than intact ones. Such a trend can be seen for MTAP levels measured from western blot – on average *MTAP*-deleted tumors have lower expression of MTAP (**Figure 2H**), although MTAP signal comes from both cancer cells and stroma. Such a trend should also be seen if *MTAP*-deleted cancer cells inside tumors express lower SDMA trend. However, we do not see any specific trend in levels of SDMA – on average *MTAP*-deleted tumors do not express lower levels of SDMA.

3. Moreover, we investigated the effects of the co-culture of *MTAP*-deleted with suspended *MTAP*-intact cells on PRMT5 activity of the *MTAP*-deleted cells. **Figure 4F** shows that co-culture of *MTAP*-deleted cancer cell lines with *MTAP*-intact myeloid cells prevents loss of SDMA, indicating maintained PRMT5 activity. *MTAP*-deleted and reconstituted glioma cells were co-cultured with myeloid leukemia cells (MV-4-11, a monomyelocytic suspension cell line). Before lysate preparation, suspended MV-4-11 myeloid cells were removed, and adherent cancer cells were extensively washed. For each independent *MTAP*-deleted cell line, SDMA levels increased with myeloid co-culture, indicating restoration of PRMT5 activity and decreasing intracellular MTA levels in *MTAP*-deleted cells. These findings emphasize that the co-culture of *MTAP*-deleted cells with *MTAP*-intact cells negates the selective vulnerability caused by *MTAP* deletion, thereby indicating that *MTAP* deletion sensitivity in a heterogeneous human GBM tumor may be attenuated by the presence of stroma in the tumor microenvironment.

Considering that the main message of the paper is based upon this experimental setting, the authors should make sure that the signal they measure in vivo is a genuine readout of the metabolite levels in cancer cells, and not a reflection of a dilution effect from the stroma. The referee is aware that this request is technically very challenging, so another possible option is to clarify the limitations of the in vivo studies and tone down the conclusions.

- ⇒ While we agree with the reviewer that it would be preferable to measure MTA in single sorted tumor vs. stromal cells, the technology for this is not yet sufficiently robust and is not widely available. Furthermore, the preparation time for single-cell sorting and analysis would invariably perturb or alter the metabolic state (e.g., the composition of cell sorting media). All in all, we do not see a technical means of further perfecting this analysis at this time, except to say stromal dilution is insufficient to account for the lack of overall MTA elevation in bulk tumors (given the high *MTAP*-deletion frequency and high levels of MTA reported in cell culture for *MTAP*-deleted cells). Indeed, the elevation of 2-HG is detectable in virtually every last resected mutant IDH glioma, even those with low cellularity.
- ⇒ To acknowledge the reviewer's concerns, we have added the following passage to the paper.

"Our study evaluated MTA in bulk GBM tumors which are a heterogeneous mixture of cancer and stromal cells. One shortcoming of performing metabolomics and western blots on such tumors is that the measured signal (e.i., MTA levels and SDMA levels) originated from cancer cells and stromal cells. Although the signal coming from cancer cells is diluted by stromal cells, significant metabolic aberrations such as elevation of 2-HG can still be seen in resected bulk tumors (**Figure 2F** and **Supplementary Figure S7**). However, our data of profiling GBM tumors from two different data set (HF series and MDA series) shows that there are nonsignificant 1.4-fold ($p=0.2$) and 1.14-fold

($p=0.9$) increase in MTA levels in *MTAP*-deleted tumors vs. intact ones. These numbers are too low to be explained by signal dilution caused by stromal contamination if cell culture data are taken as a reference point. However, our data do not rule the possibility of minor accumulation of MTA in *MTAP*-deleted glioma cells; just that such accumulation would have to be much less than reported in culture."

Reviewer #2 (Remarks to the Author):

The manuscript is improved. The author has rewritten to more precisely and accurately describe the results presented.

⇒ We thank the referee for spending time and effort to review our manuscript.

Reviewer #3 (Remarks to the Author):

Authors have responded well to queries of other reviewer.

⇒ We want to thank the reviewer for their time and effort in reviewing our manuscript.

1. Sanderson, S. M., Mikhael, P. G., Ramesh, V., Dai, Z. & Locasale, J. W. Nutrient availability shapes methionine metabolism in p16/ *MTAP* -deleted cells. *Sci. Adv.* **5**, eaav7769 (2019).
2. Mavrakis, K. J. *et al.* Disordered methionine metabolism in *MTAP/CDKN2A*-deleted cancers leads to dependence on *PRMT5*. *Science (80-.)*. **351**, 1208–1213 (2016).
3. Marjon, K. *et al.* *MTAP* Deletions in Cancer Create Vulnerability to Targeting of the *MAT2A/PRMT5/RIOK1* Axis. *Cell Rep.* **15**, 574–587 (2016).